# UOTIP: Unbalanced Optimal Transport Map for Unpaired Inverse Problems

## Abstract

We address the problem of unpaired image inverse problems, where only independent sets of noisy measurements and clean target signals are available. We propose a novel inverse problem solver based on Unbalanced Optimal Transport, called ***Unbalanced Optimal Transport Map for Inverse Problems (UOTIP)***. Our method formulates the reconstruction task—predicting clean target signals from noisy measurements—as learning a UOT Map from noisy measurement distribution to clean signal distribution by incorporating a likelihood-based cost function. By relaxing the exact marginal constraint, the UOT framework provides key advantages to our model: robustness to multi-level observation noise, adaptability to class imbalance between noisy and clean datasets, and generalizability to diverse noise-type scenarios. Moreover, with a quadratic cost formulation, our model effectively handles linear inverse problems with unknown corruption operators. Our experiments show that our model achieves state-of-the-art performance on unpaired image inverse problem benchmarks, across linear (super-resolution and Gaussian deblurring) and nonlinear (high dynamic range reconstruction and nonlinear deblurring) inverse problems.

## 1 Introduction

The ***inverse problem*** refers to the problem of recovering an unknown signal $\mathbf{x}$ from incomplete or noisy measurements $\mathbf{y}$ (Qayyum et al., 2022; Daras et al., 2024). Many scientific and engineering tasks can be formulated as inverse problems, including seismic imaging (Virieux & Operto, 2009), weather prediction (Huang et al., 2005), audio signal processing (Lemercier et al., 2024), and medical imaging (Song et al., 2021). In particular, this includes various ***image reconstruction tasks***, such as image denoising (Zhang et al., 2017) and super-resolution (Tang et al., 2024; Dong et al., 2014). Formally, for $\mathbf{x} \in \mathbb{R}^n$ and $\mathbf{y} \in \mathbb{R}^m$, the inverse problem can be expressed as follows:

$$\mathbf{y} = \mathcal{A}(\mathbf{x}) + \mathbf{n}. \tag{1}$$

where $\mathcal{A}$ denotes the (possibly nonlinear) corruption operator and $\mathbf{n}$ represents measurement noise, which is typically assumed to be Gaussian $\mathcal{N}(0, \sigma_{\mathbf{y}}^2 I_m)$. For example, in super-resolution and Gaussian deblurring tasks, the corruption operator $\mathcal{A}$ corresponds to a downsampling operator (e.g., average pooling over $4 \times 4$ patches) and a convolution operator with a Gaussian kernel, respectively.

A key difficulty of the inverse problem is *ill-posedness*, where the mapping $\mathbf{y} \mapsto \mathbf{x}$ may not have a unique solution (Engl & Ramlau, 2015). To address such challenges, a Bayesian approach incorporates prior knowledge about $\mathbf{x}$ through the prior distribution $p(\mathbf{x})$, leading to a *Maximum a Posteriori* (MAP) estimate. Given a measurement $\mathbf{y}_0$, the MAP estimate is

$$\hat{\mathbf{x}}_{\mathrm{MAP}}(\mathbf{y_0}) = \arg\max_{\mathbf{x}} p(\mathbf{x}|\mathbf{y_0}) = \arg\min_{\mathbf{x}} \left[ -\log p(\mathbf{y_0}|\mathbf{x}) - \log p(\mathbf{x}) \right]. \tag{2}$$

where $\log p(\mathbf{y}_0|\mathbf{x})$ represents the log-likelihood of the measurement $\mathbf{y}_0$ given the estimate $\mathbf{x}$. Intuitively, the MAP estimate addresses the ill-posedness of the inverse problem by selecting plausible samples $\mathbf{x}$ from the prior distribution $p(\mathbf{x})$ that also achieve a high likelihood for producing the observation $\mathbf{y}_0$. In this regard, selecting an appropriate prior distribution $p(\mathbf{x})$ for the target (clean) signal $\mathbf{x}$ is important.

Early approaches relied on hand-crafted priors, such as sparsity (Candès & Wakin, 2008), low-rank (Fazel et al., 2008), and total variation (Candès et al., 2006). However, such hand-crafted priors

often fail to effectively characterize natural (desired) signals from unnatural signals (Ulyanov et al., 2018). To overcome this limitation, generative models have emerged as powerful alternatives for representing complex natural signals, such as GANs (Wang et al., 2022; Bora et al., 2017), VAEs (Goh et al., 2019), Optimal Transport Map (Tang et al., 2024; Korotin et al., 2023), and Diffusion models (Song et al., 2021; Chung et al., 2022; Zhang et al., 2025).

We propose a novel approach that implicitly represents the prior distribution via Unbalanced Optimal Transport (UOT), a generalization of standard Optimal Transport. We refer to our model as the ***Unbalanced Optimal Transport map for Inverse Problems (UOTIP)***. Specifically, we formulate the unpaired inverse problems as learning the UOT Map from the noisy measurement distribution to the target signal distribution. Our experiments demonstrate that our model achieves state-of-the-art performance among OT-based direct transport methods on unpaired inverse problem benchmarks across linear and nonlinear inverse problems. Moreover, the UOT problem relaxes the exact marginal-matching constraint of the standard OT problem. This flexibility provides our model various advantages, such as robustness to multi-level observation noise, adaptability to class imbalance between unpaired noisy observation and clean signal data, and generalizability to diverse noise-type scenarios. These properties make our model particularly effective in real-world settings. Our contributions can be summarized as follows:

- We introduce the first model for unpaired inverse problems based on the Unbalanced Optimal Transport by introducing the likelihood cost function.
- Our model demonstrates state-of-the-art performance among OT-based direct transport methods on both linear and nonlinear unpaired inverse problem benchmarks.
- Our UOT formulation offers our model robustness to multi-level observation noise and adaptability to class imbalance, making it effective in real-world scenarios.
- With the quadratic cost, our model can handle linear inverse problems with unknown corruption operators and still achieve competitive performance.

## 2 BACKGROUND

**Notations and Assumptions** Let $\mathcal{X}$, $\mathcal{Y}$ be two compact complete metric spaces, and let $\mu$ and $\nu$ denote probability distributions on $\mathcal{Y}$ and $\mathcal{X}$, respectively. Both $\mu$ and $\nu$ are assumed to be absolutely continuous with respect to the Lebesgue measure. Throughout this work, the source distribution $\mu$ and the target distribution $\nu$ represent the distributions of noisy measurements $\mathbf{y}$ and clean target signals $\mathbf{x}$, respectively. For a measurable map $T$, $T_{\#}\mu$ represents the pushforward distribution of $\mu$. The set $\Pi(\mu, \nu)$ denote the set of joint probability distributions on $\mathcal{Y} \times \mathcal{X}$ whose marginals are $\mu$ and $\nu$, respectively. Finally, given a function $f : \mathbb{R} \to [-\infty, \infty]$, its convex conjugate $f^*$ is defined as $f^*(y) = \sup_{x \in \mathbb{R}}\{\langle x, y \rangle - f(x)\}$.

**Optimal Transport** The *Optimal Transport (OT)* problem seeks a cost-minimizing way to transport one probability distribution to another (Villani et al., 2009; Santambrogio, 2015). Formally, the OT problem aims to map a source distribution $\mu \in \mathcal{P}(\mathcal{Y})$ to a target distribution $\nu \in \mathcal{P}(\mathcal{X})$ while minimizing a given cost function $c(\cdot, \cdot)$. The Monge's OT problem (Eq. 3) involves finding a deterministic transport map $T$ such that $T_{\#}\mu = \nu$ (Monge, 1781) (Fig. 1).

$$C(\mu, \nu) := \inf_{T_{\#}\mu = \nu} \left[ \int_{\mathcal{Y}} c(\mathbf{y}, T(\mathbf{y}))d\mu(\mathbf{y}) \right]. \tag{3}$$

Intuitively, Monge's OT problem seeks an optimal transport map $T^\star$ that generates the target distribution $\nu$ by mapping each input $\mathbf{y}$ in a way that minimizes the transport cost $c(\mathbf{y}, T(\mathbf{y}))$. However, Monge's formulation is non-convex, and a deterministic optimal transport map $T^\star$ may not exist depending on the properties of $\mu$ and $\nu$ (Villani et al., 2009; Choi et al., 2025). To address these limitations, Kantorovich proposed a relaxed formulation of the OT problem (Kantorovich, 1948), which models transport via stochastic couplings $\pi$:

$$C_{ot}(\mu, \nu) := \inf_{\pi \in \Pi(\mu, \nu)} \left[ \int_{\mathcal{Y} \times \mathcal{X}} c(\mathbf{y}, \mathbf{x})d\pi(\mathbf{y}, \mathbf{x}) \right]. \tag{4}$$

Here, $\pi \in \Pi(\mu, \nu)$ denotes a coupling of $\mu$ and $\nu$, i.e., a joint distribution with marginals $\mu$ and $\nu$. Unlike Monge's OT problem, the Kantorovich formulation (Eq. 4) guarantees the existence of an

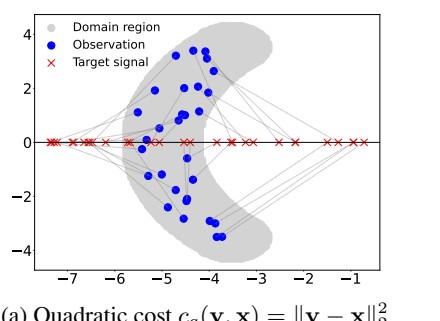 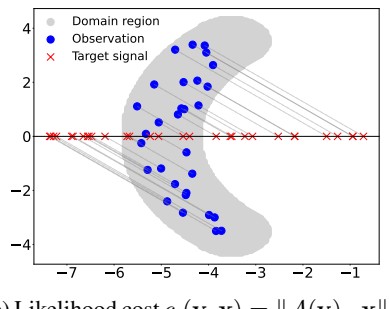

(a) Quadratic cost $c_q(\mathbf{y}, \mathbf{x}) = \|\mathbf{y} - \mathbf{x}\|_2^2$  (b) Likelihood cost $c_l(\mathbf{y}, \mathbf{x}) = \|\mathcal{A}(\mathbf{y}) - \mathbf{x}\|_2^2$.

Figure 1: **OT Maps under different cost functions** ($\mathcal{A}$: projection onto $x$-axis along $d = (1, -1)$)

optimal transport plan $\pi^\star$ under mild assumptions (Villani et al., 2009). Furthermore, if $\mu$ and $\nu$ are absolutely continuous with respect to the Lebesgue measure (our assumption), the optimal transport map $T^*$ exists and the optimal plan is induced by it, i.e., $\pi^\star = (Id \times T^\star)_{\#}\mu$ (Villani et al., 2009).

The goal of ***Neural Optimal Transport (Neural OT)*** is to learn the optimal transport map $T^\star$ using neural networks. Rout et al. (2022); Fan et al. (2023) proposed a method based on the semi-dual formulation of OT (Eq. 5), where the potential function $v_\phi$ and the transport map $T_\theta$ are both parameterized by neural networks. Intuitively, analogous to GANs, the potential function $v_\phi$ and the transport map $T_\theta$ play a similar role to the discriminator and generator.

$$\mathcal{L}_{v_\phi, T_\theta} = \sup_{v_\phi} \left[ \int_{\mathcal{Y}} \inf_{T_\theta} \left[ c\left(\mathbf{y}, T_\theta(\mathbf{y})\right) - v_\phi\left(T_\theta(\mathbf{y})\right) \right] d\mu(\mathbf{y}) + \int_{\mathcal{X}} v_\phi(\mathbf{x}) d\nu(\mathbf{x}) \right]. \quad (5)$$

**Unbalanced Optimal Transport**  The *Unbalanced Optimal Transport (UOT)* problem (Chizat et al., 2018; Liero et al., 2018) is a generalization of the standard OT problem (Eq. 4) that allows flexibility in the source distribution $\mu$ and target distribution $\nu$:

$$C_{uot}(\mu, \nu) = \inf_{\pi \in \mathcal{M}_+(\mathcal{Y} \times \mathcal{X})} \left[ \int_{\mathcal{Y} \times \mathcal{X}} c(\mathbf{y}, \mathbf{x}) d\pi(\mathbf{y}, \mathbf{x}) + D_{\Psi_1}(\pi_0 | \mu) + D_{\Psi_2}(\pi_1 | \nu) \right], \quad (6)$$

where $\mathcal{M}_+(\mathcal{Y} \times \mathcal{X})$ denotes the set of positive Radon measures on $\mathcal{Y} \times \mathcal{X}$. The terms $D_{\Psi_1}$ and $D_{\Psi_2}$ are $f$-divergences induced by convex functions $\Psi_i$, penalizing deviations of the marginals of $\pi$ from $\mu$ and $\nu$, respectively, i.e., $D_{\Psi_i}(\pi_j | \eta) = \int \Psi_i \left( \frac{d\pi_j(x)}{d\eta(x)} \right) d\eta(x)$. Note that the standard OT problem enforces exact matching of the marginals, i.e., $\pi_0 = \mu, \pi_1 = \nu$. In contrast, UOT relaxes this constraint by adding $f$-divergence penalties on the marginals. By minimizing these penalty terms, the marginals of the optimal UOT plan $\pi_u^\star$ are softly matched to $\mu$ and $\nu$, i.e., $\pi_{u,0}^\star \approx \mu$ and $\pi_{u,1}^\star \approx \nu$.

This relaxation provides UOT robustness to outliers (Balaji et al., 2020; Séjourné et al., 2022; Gazdieva et al., 2025) and the ability to handle class imbalance between marginal distributions (Eyring et al., 2024; Lee et al., 2025). Here, the class imbalance problem refers to the scenario where the source and target distributions consist of different proportions of classes. The UOT map can be interpreted as the OT Map between rescaled marginal distributions $\pi_{u,0}^\star(\mathbf{y}) = \Psi_1^{*\prime}(-v^{\star c}(\mathbf{y}))\mu(\mathbf{y})$ and $\pi_{u,1}^\star(\mathbf{x}) = \Psi_2^{*\prime}(-v^\star(\mathbf{x}))\nu(\mathbf{x})$, where $v^\star$ denotes the optimal potential (Eq. 13) (Choi et al., 2023). These rescaling factors $\Psi_i^{*\prime}$ enables the UOT Map $T_u^\star$ to naturally handle class imbalance by reweighting source samples. For example, $T_u^\star$ can make a correspondence between a mode representing 20% of the source distribution and a mode representing 30% of the target distribution.

Choi et al. (2023) introduced a Neural OT model for the UOT problem with the quadratic cost $c(\mathbf{y}, \mathbf{x}) = \|\mathbf{y} - \mathbf{x}\|_2^2$, called *UOTM*, and applied this to generative modeling. We extend the Neural UOT model to inverse problems by generalizing the cost function. In Sec 3, we demonstrate how inverse problems can be naturally formulated within the Neural OT framework and present our model.

## 3 METHOD

In this section, we present our UOT-based model for inverse problems, called ***Unbalanced Optimal Transport Map for Inverse Problems (UOTIP)***. The core idea of UOTIP is to formulate the inverse

problem solver as an Unbalanced Optimal Transport (UOT) map from the noisy measurements $\mathbf{y}$ to the target signal $\mathbf{x}$ (Eq. 1). In Sec 3.1, we introduce our UOT-based formulation of inverse problems. In Sec 3.2, we describe the learning objective, which is derived as a generalization of the vanilla Neural UOT model (Choi et al., 2023) under the proposed cost function.

## 3.1 Optimal Transport Formulation of Unpaired Inverse Problems

**Task Formulation**    In this subsection, we interpret the inverse problems solver as a Neural (Unbalanced) Optimal Transport Map $T^\star$ that maps the noisy measurements $\mathbf{y}$ to the target signal $\mathbf{x}$ under an appropriate cost function $c(\cdot, \cdot)$ (to be specified in Eq. 12).

Our main target task is the **unpaired image inverse problems with a known corruption operator** $\mathcal{A}$ **and unknown noise level** $\sigma_y > 0$ **(Eq. 1)**. (The case of an unknown corruption operator will be discussed in Sec 4.3.) Formally, let $Y = \{\mathbf{y}_i : \mathbf{y}_i \in \mathcal{Y}, i = 1, \cdots, M\}$ and $X = \{\mathbf{x}_j : \mathbf{x}_j \in \mathcal{X}, j = 1, \cdots, N\}$ denote the sets of *noisy measurements* and *target signals (clean image)*, respectively. Here, we assume the *unpaired* setting, i.e., $Y$ and $X$ are independently sampled from the source distribution $\mu$ and the target distribution $\nu$. Then, our goal is to learn an inverse problem solver $T$:

$$T : \mathcal{Y} \to \mathcal{X}, \qquad \mathbf{y} \mapsto T(\mathbf{y}) \tag{7}$$

from these unpaired training data. Following the MAP approach (Eq. 2), $T$ should satisfy:

   (i) The estimate $T(\mathbf{y})$ follows the target signal distribution, i.e., $T_{\#}\mu = \nu$.
   (ii) For each given $\mathbf{y}_0$, $T(\mathbf{y}_0)$ maximizes the log-likelihood $\log p(\mathbf{y}_0|\cdot)$.

Here, the first condition (i) represents that the predicted target signal $T(\mathbf{y})$ should maintain higher fidelity with respect to the target distribution $\nu$. In this regard, (i) can be relaxed by instead requiring

   (i-1) $T(\mathbf{y})$ attains a high likelihood under the target signal distribution, i.e., $\nu(T(\mathbf{y})) > M$ for all $\mathbf{y}$ for some threshold $M > 0$.

Our main observation is that **these two conditions can be naturally interpreted through the (Unbalanced) Optimal Transport** (Eq. 3 and 6). Formally, in the OT problem, the OT map $T^\star$ is defined as the transport cost minimizer over valid transport maps:

   (a) Valid transport maps satisfying $T^\star_{\#}\mu = \nu$.
   (b) Each $\mathbf{y}$ is mapped to minimize the transport cost $\int_{\mathcal{Y}} c(\mathbf{y}, T^\star(\mathbf{y}))d\mu(\mathbf{y})$.

Therefore, $T^\star$ naturally satisfies the first condition (i) of the inverse problem solver. **Our approach is to design an appropriate cost function** $c(\cdot, \cdot)$ **so that the second condition (ii) is also satisfied**. Under this formulation, the Neural OT model can effectively serve as the inverse problem solver.

**Likelihood Cost and MAP Estimate**    Our goal is to solve inverse problems using a Neural OT model. To adapt the general OT map as an inverse problem solver, we introduce the ***likelihood cost function***, which can be adapted to each specific inverse problem.

$$c_l(\mathbf{y}, \mathbf{x}) = \|\mathcal{A}(\mathbf{x}) - \mathbf{y}\|_2^2 \tag{8}$$

Note that under Gaussian measurement noise, $c_l$ is proportional to the negative log-likelihood of the observation $-\log p(\mathbf{y} \mid \mathbf{x})$. Fig. 1 illustrates how the OT map $T^\star$ changes under the likelihood cost and the quadratic cost. In addition, in Sec 4.2, we will show experimentally that this cost function can be applied to other noise types as well.

Interestingly, the Neural OT model with the likelihood cost admits a MAP interpretation. In particular, adopting the cost function $c(\mathbf{y}, \mathbf{x}) = -\log p(\mathbf{y}|\mathbf{x})$ is equivalent to minimizing the negative log-posterior $-\log p(\mathbf{x}|\mathbf{y})$, within the OT framework:

$$C_{ot}(\mu, \nu) = \inf_{\pi \in \Pi(\mu,\nu)} \left[ \int_{\mathcal{Y} \times \mathcal{X}} -\log p(\mathbf{x}|\mathbf{y})d\pi(\mathbf{y}, \mathbf{x}) \right]. \tag{9}$$

$$= \inf_{\pi \in \Pi(\mu,\nu)} \left[ \int_{\mathcal{Y} \times \mathcal{X}} -\log p(\mathbf{y}|\mathbf{x}) - \log p(\mathbf{x}) + \log p(\mathbf{y})d\pi(\mathbf{y}, \mathbf{x}) \right]. \tag{10}$$

$$= \inf_{\pi \in \Pi(\mu,\nu)} \left[ \int_{\mathcal{Y} \times \mathcal{X}} -\log p(\mathbf{y}|\mathbf{x})d\pi(\mathbf{y}, \mathbf{x}) \right]. \tag{11}$$

The last equality holds because, for $\pi \in \Pi(\mu, \nu)$, the marginal distributions are fixed to $\pi_0 = \mu$ and $\pi_1 = \nu$. This result shows that, under likelihood cost, **the OT formulation of inverse problems implicitly incorporates the target signal prior** $p(\mathbf{x})$ through the valid transport map conditions, i.e., $T_{\#}\mu = \nu$ or $\pi \in \Pi(\mu, \nu)$.

**Quadratic Cost Benefits** To complement the likelihood cost $c_l$, we also include the quadratic cost $c_q$. The resulting overall cost function is expressed as follows:

$$c(\mathbf{y}, \mathbf{x}) = \tau \left( c_l(\mathbf{y}, \mathbf{x}) + c_q(\mathbf{y}, \mathbf{x}) \right) \quad \text{where } c_l(\mathbf{y}, \mathbf{x}) = \|\mathcal{A}(\mathbf{x}) - \mathbf{y}\|_2^2 \text{ and } c_q(\mathbf{y}, \mathbf{x}) = \|\mathbf{y} - \mathbf{x}\|_2^2. \tag{12}$$

Here, $\tau$ denotes the cost intensity parameter. This additional quadratic term serves two purposes. First, it provides favorable theoretical guarantees. Inverse problems are typically ill-posed. From the OT perspective, this ill-posedness can cause the likelihood cost $c_l$ to violate the *twist condition*, which is critical for guaranteeing the existence of the OT Map $T^\star$. Under mild assumptions, incorporating the quadratic cost $c_q$ resolves this issue by ensuring the overall cost function $c(\cdot, \cdot)$ satisfies the twist condition (See Appendix C for details.). In practice, we additionally require the forward operator $\mathcal{A}$ to be differentiable because the gradient of $\mathcal{A}$ is required in the optimization process (Algorithm 1).

**Remark 1.** *For ill-posed inverse problems such as Gaussian deblurring or HDR reconstruction, the additional quadratic cost $c_q$ ensures that our overall cost function $c(\cdot, \cdot)$ (Eq. 12) satisfies the twist condition, which guarantees the existence of the OT map. Formally, if $\mathcal{A}$ is $L$-Lipschitz continuous, then $c_l(\mathbf{y}, \mathbf{x}) + \lambda c_q(\mathbf{y}, \mathbf{x})$ satisfies the twist condition when $\lambda > L$.*

Second, from a practical perspective, this quadratic cost allows us to extend our model to scenarios with an unknown corruption operator (by using quadratic cost only). As shown in the ablation study in Sec 4.3, our quadratic-only variant still achieves competitive performance on linear inverse problems even without explicit knowledge of the corruption operator $\mathcal{A}$.

## 3.2 PROPOSED MODEL

In this subsection, we present the learning objective of our UOTIP model and then highlight the theoretical advantages of adopting the UOT map over the standard OT map.

**Learning Objective** Our approach is to learn the UOT Map $T_u^\star$ from the noisy measurement distribution $\mu$ to the target signal distribution $\nu$ using a neural network $T_\theta$. To this end, we employ the UOTM framework (Choi et al., 2023). This approach is based on the semi-dual formulation of the Unbalanced Optimal Transport (Vacher & Vialard, 2023) (Eq. 13).

$$C_{uot}(\mu, \nu) = \sup_{v \in \mathcal{C}} \left[ \int_{\mathcal{Y}} -\Psi_1^* \left( -v^c(\mathbf{y}) \right) d\mu(\mathbf{y}) + \int_{\mathcal{X}} -\Psi_2^* (-v(\mathbf{x})) d\nu(\mathbf{x}) \right], \tag{13}$$

where $v^c(\mathbf{y})$ denotes the $c$-transform of $v$, which is defined as $v^c(\mathbf{y}) = \inf_{\mathbf{x} \in \mathcal{X}} \left( c(\mathbf{y}, \mathbf{x}) - v(\mathbf{x}) \right)$. We refer to the $v \in \mathcal{C}$ as the potential function. Then, $T_\theta$ is parameterized to approximate the UOT Map $T_u^\star$ as follows:

$$T_\theta(\mathbf{y}) \in \underset{\mathbf{x} \in \mathcal{X}}{\operatorname{arginf}} \left[ c(\mathbf{y}, \mathbf{x}) - v(\mathbf{x}) \right] \quad \Leftrightarrow \quad v^c(\mathbf{y}) = c \left( \mathbf{y}, T_\theta(\mathbf{y}) \right) - v \left( T_\theta(\mathbf{y}) \right), \tag{14}$$

Note that this parameterization leverages the optimality condition, which is satisfied by the UOT Map $T_u^\star$ and the optimal potential function $v^\star$ (Choi et al., 2023). Finally, by substituting $v^c$ from Eq. 13 with Eq. 14 and parameterizing the potential function as $v_\phi$ with neural network, we obtain the following learning objective with the cost function from Eq. 12:

$$\mathcal{L}_{v_\phi, T_\theta} = \inf_{v_\phi} \int_{\mathcal{Y}} \Psi_1^* \left( -\inf_{T_\theta} \left[ c \left( \mathbf{y}, T_\theta(\mathbf{y}) \right) - v_\phi \left( T_\theta(\mathbf{y}) \right) \right] \right) d\mu(\mathbf{y}) + \int_{\mathcal{X}} \Psi_2^* \left( -v_\phi(\mathbf{x}) \right) d\nu(\mathbf{x}). \tag{15}$$

Here, the convex conjugates $\Psi_1^*$ and $\Psi_2^*$ are monotone increasing convex functions, determined by the marginal penalty terms in the UOT problem (Eq. 6). For example, if $D_{\Psi_i}$ is the KL divergence, then $\Psi_i^*(\cdot) = \exp(\cdot) - 1$. Moreover, our UOT map objective reduces to the standard OT map variant (Eq. 5) when $\Psi_i^*(\cdot) = Identity(\cdot)$, which corresponds to selecting $\Psi_i$ as the convex indicator function at $\{1\}$. Hence, **the Neural UOT can be interpreted as a generalization of the Neural OT**. For the training algorithm, refer to Algorithm 1.

Table 1: **Quantitative results on unpaired inverse problems**: two linear (Gaussian deblurring, Super-resolution) and two nonlinear (HDR Reconstruction, Nonlinear Deblurring). The **boldface** and underlined values indicate the best and second-best performance. Our model consistently achieves strong performance, attaining the best scores in nearly all cases.

| Task | Method | FFHQ | | | | AFHQ | | | |
|---|---|---|---|---|---|---|---|---|---|
| | | PSNR ($\uparrow$) | SSIM ($\uparrow$) | LPIPS ($\downarrow$) | FID ($\downarrow$) | PSNR ($\uparrow$) | SSIM ($\uparrow$) | LPIPS ($\downarrow$) | FID ($\downarrow$) |
| Gaussian Deblurring | NOT (Korotin et al., 2023) | 20.11 | 0.6035 | 0.209 | 52.901 | 19.99 | 0.5472 | 0.273 | 58.927 |
| | OTUR (Wang et al., 2022) | 23.82 | 0.7106 | 0.136 | 24.337 | 23.91 | 0.6777 | 0.165 | 30.773 |
| | RCOT (Tang et al., 2024) | 22.07 | 0.5492 | 0.279 | 123.692 | 22.34 | 0.5365 | 0.268 | 132.465 |
| | UOTIP (Ours) | **24.06** | **0.7139** | **0.124** | **21.210** | **24.22** | **0.6804** | **0.139** | **12.566** |
| Super-resolution $4\times$ | NOT (Korotin et al., 2023) | 20.13 | 0.6257 | 0.209 | 50.066 | 20.14 | 0.5833 | 0.261 | 44.252 |
| | OTUR (Wang et al., 2022) | 24.09 | 0.7243 | 0.129 | 22.751 | 24.71 | 0.7079 | 0.148 | 19.575 |
| | RCOT (Tang et al., 2024) | 24.05 | 0.6820 | 0.260 | 118.776 | **25.04** | 0.7137 | 0.208 | 69.072 |
| | UOTIP (Ours) | **24.35** | **0.7371** | **0.118** | **19.475** | 24.97 | **0.7142** | **0.133** | **15.939** |
| HDR Reconstruction | NOT (Korotin et al., 2023) | 21.24 | 0.7978 | 0.138 | 25.842 | 23.36 | 0.8179 | 0.127 | 10.528 |
| | OTUR (Wang et al., 2022) | 25.32 | 0.8545 | 0.076 | **16.458** | 26.25 | 0.8542 | 0.091 | **7.227** |
| | RCOT (Tang et al., 2024) | 19.26 | 0.6755 | 0.133 | 33.422 | 18.99 | 0.7060 | 0.166 | 27.767 |
| | UOTIP (Ours) | **26.02** | **0.8642** | **0.064** | 20.840 | **26.40** | **0.8653** | **0.074** | 7.897 |
| Nonlinear Deblurring | NOT (Korotin et al., 2023) | 21.37 | 0.7373 | 0.157 | 43.661 | 23.03 | 0.7271 | 0.158 | 17.377 |
| | OTUR (Wang et al., 2022) | 26.94 | 0.8594 | 0.068 | 12.538 | 26.09 | 0.8253 | 0.092 | 7.651 |
| | RCOT (Tang et al., 2024) | 25.14 | 0.7221 | 0.151 | 52.268 | 24.48 | 0.7172 | 0.153 | 29.902 |
| | UOTIP (Ours) | **28.52** | **0.8841** | **0.051** | **11.370** | **27.74** | **0.8589** | **0.069** | **5.113** |

**Unbalanced OT vs. OT**   Our goal is to solve inverse problems under an unpaired setting, where the training datasets $Y$ and $X$ are not given as paired samples. The UOT (Eq. 6) is a generalization of the OT by relaxing the marginal distribution constraints. This flexibility provides several distinctive advantages to the UOT map $T_u^\star$:

(a$^*$)  Greater flexibility in matching target distributions (Eq. 6).

(b$^*$)  The ability to address class imbalance between the source and target distributions (Eyring et al., 2024; Lee et al., 2025).

(c$^*$)  In Neural OT models, more stable training dynamics and thereby better fidelity to the target distribution compared to the OT Map (Choi et al., 2023; 2024).

These properties make the UOT map particularly well-suited for our unpaired inverse problem setting. Because the training datasets $Y$ and $X$ are not given in one-to-one correspondence, enforcing strict marginal matching, i.e., $T_\#\mu = \nu$, as in (i), can be too restrictive. Instead, the relaxed constraint (a$^*$) enables the UOT Map $T_u^\star$ to satisfy the softer condition (i-1) in Sec 3.1, by focusing more on high-likelihood regions. To be more specific, this flexibility allows small marginal mismatches by prioritizing majority modes while reducing outliers (Balaji et al., 2020; Séjourné et al., 2022). Moreover, (b$^*$) provides $T_u^\star$ the ability to handle the case with multiple measurement noise levels $\sigma_y$, which will be evaluated in Sec 4.2. In such cases, several noisy measurements **y** may correspond to the same target signal **x**. In this regard, this scenario can be interpreted as one instance of class imbalance. Lastly, (c$^*$) improves training stability by upper-bounding the gradient norm of potential functions (Choi et al., 2024). Given these benefits, our model is built upon the UOT Map.

# 4 EXPERIMENTS

In this section, we evaluate the performance of our model from diverse perspectives.

- In Sec 4.1, we assess our model on four inverse problem benchmarks, including linear inverse problems and nonlinear inverse problems.
- In Sec 4.2, we investigate several advantages of our model, derived from the Unbalanced Optimal Transport formulation, such as handling multi-noise level observations, addressing the class imbalance problem, and managing diverse noise types.
- In Sec 4.3, we conduct an ablation study on the likelihood cost function $c_l$ (See Appendix F.1 for additional results on the cost intensity hyperparameter $\tau$).

## 4.1 UNPAIRED INVERSE PROBLEM BENCHMARKS

**Setting**   We evaluated our model on four unpaired inverse problem benchmarks: *two linear (Gaussian deblurring, super-resolution)* and *two nonlinear (HDR reconstruction, nonlinear deblurring)*,

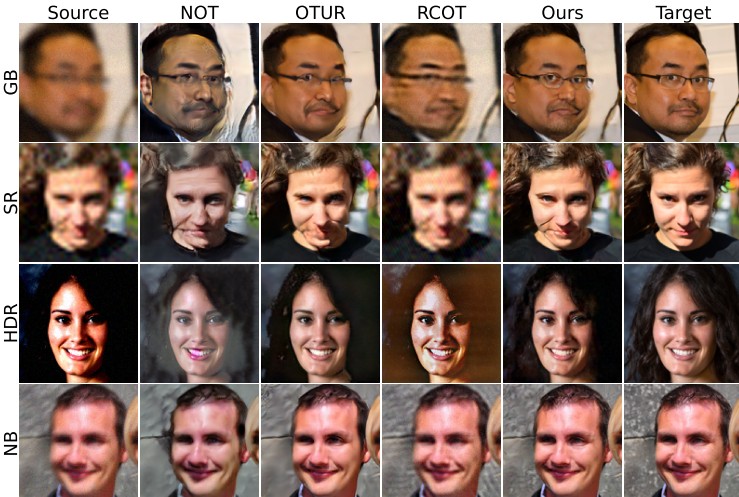

Figure 2: **Comparison of inverse problem solvers on FFHQ** for four tasks: Gaussian deblurring (GB), Super-resolution (SR), High dynamic range reconstruction (HDR), and Nonlinear deblurring (NB). Our model produces higher fidelity images with well-preserved textures.

with additive white Gaussian noise ($\sigma_{\mathbf{y}} = 0.05$). For the super-resolution, the standard quadratic cost $c_q(\mathbf{y}, \mathbf{x})$ cannot be directly applied because $\mathbf{y}$ and $\mathbf{x}$ have different dimensions. Hence, we introduce bicubic interpolation $Q$ on the lower-resolution image $\mathbf{y}$ and define the modified cost as $c_q(\mathbf{y}, \mathbf{x}) = \|Q(\mathbf{y}) - \mathbf{x}\|_2^2$.[1] In the unpaired setting, the training datasets of measurements $Y = \{\mathbf{y}\}$ and target signals $X = \{\mathbf{x}\}$ are not given in pairs. During training, $\mathbf{y}$ and $\mathbf{x}$ are independently sampled from $Y_{train}$ and $X_{train}$, respectively. Implementation details are provided in Appendix B.

We test our model on two image datasets: **FFHQ** (Karras et al., 2019) and **AFHQ-dog** (Choi et al., 2020), resized to $128 \times 128$. For quantitative evaluation, we employ PSNR and SSIM for assessing pixel-level similarity with the ground-truth target signal, and LPIPS and FID for perceptual quality. We compare our model against existing approaches for unpaired inverse problems: GAN-based *OTUR (Wang et al., 2022)* and OT-map–based *NOT (Korotin et al., 2023)* and *RCOT (Tang et al., 2024)*).

**Experimental Results**    Table 1 shows the quantitative results on four inverse problems across two datasets. Our UOTIP achieves the best performance on almost all metrics across inverse problems and datasets. Specifically, in Gaussian deblurring and nonlinear deblurring, our method consistently outperforms all other approaches across all four metrics on both datasets. In super-resolution, our method achieves the best scores except for PSNR on AFHQ, where RCOT attains a comparable PSNR score to ours. Here, our approach attains significantly better FID scores on both datasets, indicating that our method attains superior fidelity of target signals compared to other methods.

Fig. 2 shows qualitative comparisons on the FFHQ dataset (see Figure 7 in the Appendix E.1 for AFHQ results and Appendix E.2 for additional examples). The results further demonstrate our superior image fidelity. NOT often overemphasize features, such as contours and textures, while RCOT fails to reliably remove degradations. OTUR effectively restores degradations but often distorts or over-smooths fine details, including eyes, mouths, and textural details. In contrast, our model produces sharper images with well-preserved textures.

## 4.2 Investigating Advantages from the (U)OT Framework

In this subsection, we examine the benefits of our UOT formulation for unpaired inverse problems. Specifically, our UOTIP is tested under ***multi-level noise observations, class imbalance settings,*** and ***various noise types***. The first two advantages stem from the unbalancedness of our UOT formulation. As discussed in Sec. 3.2, unlike the standard OT Map, the UOT Map $T_u^{\star}$ allows rescaling of each

---

[1]Note that under this modified cost, the twist condition in Remark 1 is no longer satisfied. Nevertheless, as shown in Table 1, UOTIP performs well on super-resolution in practice. We attribute this empirical success to the local smoothing inductive bias of the generator network architectures.

Table 2: **Quantitative results under multi-level observation noise** for linear inverse problems on FFHQ. Our model exhibits superior robustness across noise levels.

| Method | Gaussian Deblurring | | | | Super Resolution 4× | | | |
|---|---|---|---|---|---|---|---|---|
| | PSNR ($\uparrow$) | SSIM ($\uparrow$) | LPIPS ($\downarrow$) | FID ($\downarrow$) | PSNR ($\uparrow$) | SSIM ($\uparrow$) | LPIPS ($\downarrow$) | FID ($\downarrow$) |
| NOT (Korotin et al., 2023) | 19.07 | 0.5491 | 0.250 | 98.558 | 18.91 | 0.5653 | 0.264 | 99.053 |
| OTUR (Wang et al., 2022) | 22.55 | 0.6417 | 0.174 | 67.323 | 23.20 | 0.6681 | 0.178 | 70.223 |
| OTIP (Ours-OT) | 22.87 | 0.6562 | 0.160 | 91.309 | 23.21 | 0.6732 | 0.170 | 85.674 |
| UOTIP (Ours) | **23.04** | **0.6649** | **0.154** | **65.664** | **23.30** | **0.6864** | **0.157** | **58.406** |

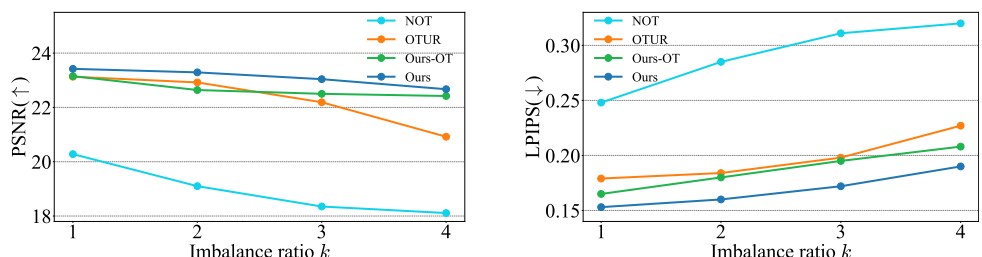

Figure 3: **Quantitative results under class imbalance** with imbalance ratio $k$ between AFHQ-cat and AFHQ-dog for the Gaussian deblurring (cat:dog $1 : 1 \rightarrow k : 1$). Our method achieves superior performance and demonstrates greater robustness across various ratios.

sample $T_u^\star(\mathbf{y})$. This flexibility provides robustness to handle varying noise levels and class imbalance in inverse problems. For the last advantage, we evaluate whether our model can generalize to various noise types. This robustness arises from the OT formulation itself.

**Multi-level Observation Noise** Most existing unpaired inverse problem solvers (Tang et al., 2024; Wang et al., 2022) assume a single fixed noise level $\sigma_{\mathbf{y}} > 0$. However, in realistic scenarios, observation noise can vary across samples. To test robustness under such conditions, we design experiments on the FFHQ dataset where noise is drawn from a mixture of four levels: $\sigma_{\mathbf{y}} \in \{0.025, 0.05, 0.1, 0.2\}$, instead of fixing $\sigma_{\mathbf{y}} = 0.05$. For each level, degraded images are independently sampled in proportions 4:3:2:1, so that multiple noisy observations $\mathbf{y} \sim \mu$ may correspond to the same clean target (Fig. 5). This setup evaluates whether our UOT-based formulation can effectively handle heterogeneous noise distributions. We compare UOTIP against *OTUR* (best baseline in Sec 4.1), *NOT* (OT-map model), and an OT variant of our model (*Ours-OT*), obtained by setting $\Psi_1(\cdot) = \Psi_2(\cdot) = \text{Identity}(\cdot)$ in Eq. 15. Since robustness to multi-level noise arises only from the unbalanced formulation, for completeness, we also compare with the OT-variant of our model.

Table 2 presents results on linear inverse problems (see Table 5 in Appendix F.2 for nonlinear results and Appendix E.3 for qualitative examples). Our UOTIP achieves the best performance on all metrics. Specifically, our model outperforms OTUR (the strongest baseline in standard benchmarks) and the OT-variant of our model (OTIP). In particular, UOTIP shows a clear improvement in FID over OTIP, highlighting its superior perceptual fidelity.

**Class Imbalance** We further evaluate UOTIP under class imbalance, where source and target distributions are multi-modal but differ in their mode proportions. This situation can naturally occur in unpaired settings when measurements and clean signals are obtained from different data sources. For example, in super-resolution, large amounts of low-resolution data may be available, whereas high-resolution data may be limited. In this case, additional public high-resolution datasets may then be incorporated, leading class imbalance between modes. To examine this, we constructed datasets by combining AFHQ-cat with AFHQ-dog (Choi et al., 2020). For each imbalance ratio $k \in \{1, 2, 3, 4\}$, we construct a target dataset $X$ of 6,000 images by mixing cat and dog samples at a $k : 1$ (cat: dog) ratio. The source dataset $Y$ is fixed at a $1 : 1$ class ratio by downsampling cat images from $X$, and is then generated by applying the degradation/measurement operator to this subset.

Fig 3 presents the quantitative results of PSNR and LPIPS metric. For clarity, we visualized only two metrics, because the two pixel-level measures (PSNR, SSIM) and the two perceptual measures (LPIPS, FID) exhibited similar trends (See Table 6 Appendix F.2 for the complete table). Similar to the multi-level noise experiments, our UOT-based model consistently outperforms the baselines under

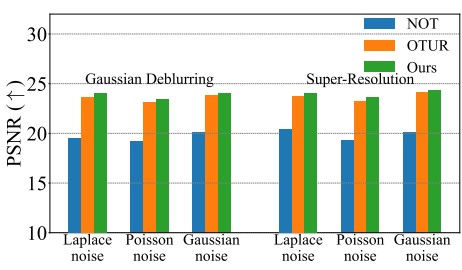 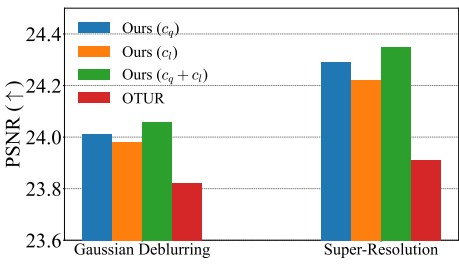

(a) Generalization to diverse noise types     (b) Ablation study on cost functions

Figure 4: **PSNR ($\uparrow$) results for diverse noise types and cost ablation.** Complete results on all four metrics are provided in Appendix F.2. In Fig.4a, our model generalizes well across different noise types. In Fig.4b, the full model achieves the best overall performance, and even the blind-corruption variant (using only $c_q$) outperforms the previous state-of-the-art OTUR model.

class imbalance, achieving the best or at least comparable performance across all metrics. OTUR and Ours-OT achieve intermediate performance, whereas NOT consistently yields the lowest results.

**Diverse Noise Types**    We incorporate the likelihood cost $c_l(\mathbf{y}, \mathbf{x})$, which is based on the negative log-likelihood of the Gaussian measurement noise, to guide the UOT map as our inverse problem solver. In this section, we examine the robustness of our model under noise-type mismatch by testing on alternative noise distributions while keeping the likelihood cost $c_l$ fixed as Gaussian. Intuitively, the UOT (or OT) map (Eq. 6) is defined as the transport cost minimizer (Condition (b)), i.e., enforcing $\mathcal{A}(T(\mathbf{y})) \approx \mathbf{y}$, among the valid transport maps (Condition (a)) in Sec 3.1. In this respect, minimizing $c_l$ is expected to yield robustness across noise types, since enforcing $\mathcal{A}(T(\mathbf{y})) \approx \mathbf{y}$ does not strictly depend on the Gaussian assumption. To validate this generalization, we test our model under two alternative noise distributions: *Laplace noise* and *Poisson noise* (See Appendix B for details).

Fig. 4a presents PSNR results on linear inverse problems on the FFHQ dataset. Due to page constraints, only PSNR is shown here; complete results across all four metrics are provided in Table 7 of Appendix F.2. Our model attains strong performance across various noise types compared to other existing approaches: NOT (Korotin et al., 2023) and OTUR (Wang et al., 2022). These results demonstrate the robustness of our approach in handling diverse noise conditions.

### 4.3    Ablation Study on Likelihood Cost $c_l$

We conduct an ablation study on our cost function $c(\cdot, \cdot)$ (Eq. 12). Note that our cost consists of two terms: the *problem-agnostic quadratic cost $c_q$* and the *problem-adaptive likelihood cost $c_l$*. We analyze the contribution of each term by excluding each term from our cost function. The PSNR results on linear inverse problem are provided in Fig. 4b (See Table 8 in Appendix F.2 for full metric table). Our full model with both cost terms achieves the best results across most quantitative metrics. This result shows that the theoretical benefit of including the quadratic cost—namely, guaranteeing the existence of a UOT map (Remark 1)—also leads to practical performance improvement. Interestingly, the quadratic-cost-only variant remains competitive and even surpasses OTUR (Wang et al., 2022). This suggests that our quadratic-cost-only formulation has potential as an effective blind inverse problem solver when the corruption operator $A$ preserves the signal structure, such as in Gaussian deblurring.

## 5    Conclusion

In this paper, we proposed UOTIP, an unpaired inverse problem solver model based on the Unbalanced Optimal Transport formulation. We formulated the inverse problem as learning the (Unbalanced) OT Map between the distributions of noisy measurements and target signals, and introduced a novel likelihood cost function. Our model achieves competitive performance on unpaired inverse problems, outperforming existing approaches. Moreover, our Unbalanced OT formulation provides key advantages over the standard OT formulation, such as robustness to multi-level observation noise and class imbalance. One limitation of our work is that our method is only tested under fixed-form cost functions, without any training capacity. While our Gaussian-based likelihood cost showed some generalization to other noise types, a broader cost design could further improve performance.

**Reproducibility Statement**    We provide the implementation code in the supplementary material, and essential details for reproducibility are included in Appendix B.

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

## A   USE OF LARGE LANGUAGE MODELS

**Paper Writing**   We use Large Language Models (LLMs) to aid or polish writing.

**Research Assistant**   We use LLMs as an auxiliary tool for code-level debugging and minor modifications.

## B   IMPLEMENTATION DETAILS

All models introduced in this paper are trained for 60,000 iterations, and we report the best results with respect to LPIPS and FID, even if they occur at intermediate iterations.

**Ours**   In our model, unless otherwise specified, the settings follow those of UOTM (Choi et al., 2023) on CelebA-256. Our framework jointly learns the potential $v_\phi$ and the Optimal Transport Map $T_\theta$. For the potential $v_\phi$, we adopt the potential architecture from UOTM (Choi et al., 2023), while for OT Map $T_\theta$, we employ the generator architecture from OTUR (Wang et al., 2022). The learning rate for the potential is $lr_{v_\phi} = 5.0 \times 10^{-5}$, and the learning rate for the OT Map is $lr_{T_\theta} = 1.0 \times 10^{-4}$. The cost intensity hyperparameter $\tau$ is fixed to 0.001. The batch size during training is fixed at 32. The convex conjugate $\Psi_i^*$ is derived from the generator function of the KL divergence,

$$\Psi(x) = \begin{cases} x \log x - x + 1, & \text{if } x > 0 \\ \infty, & \text{if } x \leq 0 \end{cases}, \tag{16}$$

which defines the associated $f$-divergence and yields the explicit form $\Psi_i^*(t) = e^t - 1$. In the case of Ours-OT, we instead set $\Psi_i^*(\cdot) = Identity(\cdot)$, while keeping all other configurations identical to those of Ours.

**Baselines**   For NOT (Korotin et al., 2023), we employ the generator and discriminator of UOTM and adopt its hyperparameter settings. The batch size is set to 32. For OTUR (Wang et al., 2022) and RCOT (Tang et al., 2024), we strictly follow all configurations as proposed in their original models.

**Dataset and Evaluation**   For FFHQ, we use 6,000 images from the original dataset as the clean signal, allocating 5,500 to the training set and 500 to the test set. For AFHQ, we use the original training set and the original test set as the clean signal. The measurements are generated by applying the forward operator (Eq. 1) to these clean signals. Note that under the unpaired assumption, minibatches of measurement $Y$ and clean signal $X$ are always independently sampled in Algorithm 1. For evaluation, PSNR, SSIM, and LPIPS are computed on the test set, whereas FID is evaluated using both the training and test sets.

**Inverse Problem**   We follow the experimental settings of (Kawar et al., 2022; Song et al., 2023) for super-resolution and (Zhang et al., 2025) for the other three tasks. Formally,

- **Gaussian deblurring** (kernel size $61 \times 61$ and kernel standard deviation 3.0):

$$\mathbf{y} = \mathbf{k} * \mathbf{x} + \mathbf{n}, \ \mathbf{n} \sim \mathcal{N}(\mathbf{0}, \sigma_y^2 \mathbf{I}_m) \tag{17}$$

   where $k$ is the Gaussian kernel and $*$ denotes the convolution operator.

- **Super-resolution** ($4 \times 4$ patch downsampling):

$$\mathbf{y} = \mathbf{x} \downarrow_4 + \mathbf{n}, \ \mathbf{n} \sim \mathcal{N}(\mathbf{0}, \sigma_y^2 \mathbf{I}_m) \tag{18}$$

- **High Dynamic Range (HDR) reconstruction** (scale factor 2.0, clipping to $[-1, 1]$):

$$\mathbf{y} = \text{clip}(2\mathbf{x}, -1, 1) + \mathbf{n}, \ \mathbf{n} \sim \mathcal{N}(\mathbf{0}, \sigma_y^2 \mathbf{I}_m) \tag{19}$$

- **Nonlinear deblurring** ($\mathcal{A}$: neural operator from (Tran et al., 2021)):

$$\mathbf{y} = \mathcal{A}(\mathbf{x}) + \mathbf{n}, \ \mathbf{n} \sim \mathcal{N}(\mathbf{0}, \sigma_y^2 \mathbf{I}_m) \ \text{ for pretrained operator } \mathcal{A} \tag{20}$$

   Here, $\mathcal{A}$ is a pretrained neural operator model on the GoPro dataset (Nah et al., 2017), which learns to approximate the nonlinear blur characteristics observed in the dataset (Tran et al., 2021).

**Multi-level Observation Noise Dataset**    We construct a multi-level observation noise dataset from FFHQ by first applying a corruption operator and subsequently adding noise drawn from a mixture of four levels: $\sigma_{\mathbf{y}} \in \{0.025, 0.05, 0.1, 0.2\}$. For each noise level, we randomly and independently sample degraded images in proportions of $4 : 3 : 2 : 1$. For evaluation, we generate a test set by applying all four noise levels to each of 500 FFHQ images, resulting in 2,000 degraded samples.

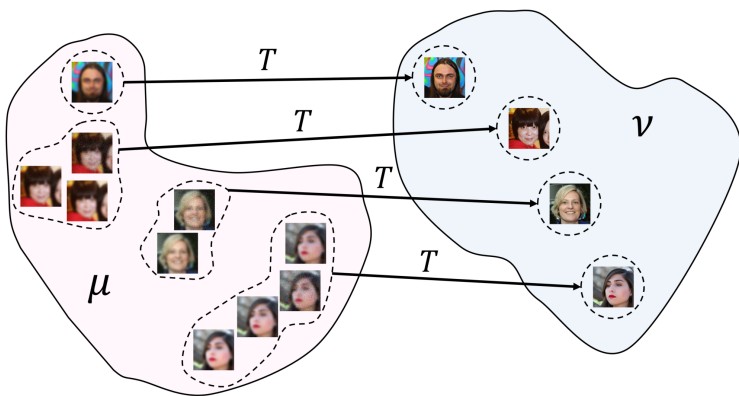

Figure 5: **Conceptual illustration of the multi-level observation noise problem**. A single target signal $\mathbf{x}$ may be observed under multiple Gaussian noise levels. Unlike the OT map, the UOT map can rescale each sample, enabling the alignment of multiple noisy observations with a single target signal. This flexibility makes our UOT-based model more robust to multi-level observation noise.

**Diverse Noise Types Setting**    For Laplace noise, the scale parameter is chosen such that its variance matches that of the Gaussian noise with $\sigma_{\mathbf{y}} = 0.05$. For Poisson noise, we follow the experimental setup described in (Chung et al., 2022). Formally,

- **Laplace noise** (Laplace noise scale $b = \frac{0.05}{\sqrt{2}}$) :

$$\mathbf{y} = \mathcal{A}(\mathbf{x}) + \mathbf{n}, \ \ \mathbf{n} \sim \text{Laplace}(\mathbf{0}, b) \tag{21}$$

- **Poisson noise** (Poisson noise level $\lambda = 1.0$):

$$p(\mathbf{y}|\mathbf{x}_0) = \prod_{j=1}^{n} \frac{[A(\mathbf{x}_0)]_j^{y_j} \exp\{-[A(\mathbf{x}_0)]_j\}}{y_j!}, \tag{22}$$

where $j$ indexes the measurement bin, i.e., $j \in \{0, 1, \dots, 255\}$. To be more specific, the Poisson noise is defined on integer pixel values $[0, 255]$. Thus, each normalized image (range $[-1, 1]$) is converted to 8-bit $[0, 255]$, Poisson noise is applied, and the result is rescaled back to $[-1, 1]$ to form the measurements.

**Algorithm**    Algorithm 1 describes the training process used in UOTIP. The generator $T_\theta$ learns to transport noisy measurements toward the target distribution by minimizing the cost term, while the discriminator $v_\phi$ is trained using the convex conjugate $\Psi_i^*(x) = e^x - 1$. By alternating these updates, the model progressively refines the transport map.

**Algorithm 1** Training algorithm of UOTIP

---

**Require:** The noisy measurement distribution $\mu$. The target image distribution $\nu$. $\Psi_i^*(t) = e^t - 1$.
Generator network $T_\theta$ and the discriminator network $v_\phi$. Total iteration number $K$.
1: **for** $k = 0, 1, 2, \ldots, K$ **do**
2:     Sample a batch $Y \sim \mu$, $X \sim \nu$.
3:     $\mathcal{L}_T = \frac{1}{|Y|} \sum_{y \in Y} c\left(y, T_\theta(y)\right) - v_\phi\left(T_\theta(y)\right)$
4:     Update $\theta$ by minimizing the loss $\mathcal{L}_T$.
5:     $\mathcal{L}_v = \frac{1}{|Y|} \sum_{y \in Y} \Psi_1^*\left(-c\left(y, T_\theta(y)\right) + v_\phi\left(T_\theta(y)\right)\right) + \frac{1}{|X|} \sum_{x \in X} \Psi_2^*(-v_\phi(x))$
6:     Update $\phi$ by minimizing the loss $\mathcal{L}_v$.
7: **end for**

---

**Training Dynamics**    We could verify our model shows the desirable training dynamics on HDR and Nonlinear blur.

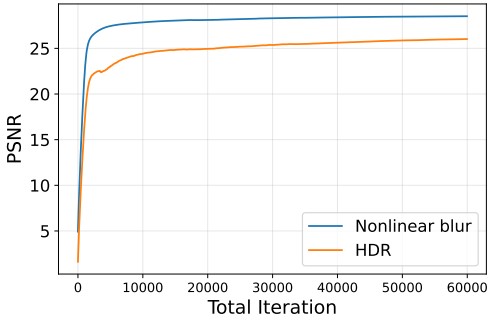 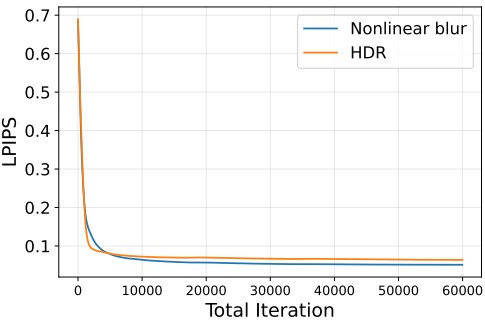

Figure 6: Training Dynamics.

**Efficiency Analysis**    Table 3 shows UOTIP achieves substantially lower training time than RCOT while maintaining reasonable parameter size. Compared to NOT and RCOT, UOTIP is both more computationally efficient and more compact, demonstrating clear advantages in practical deployment.

| Training (60,000 iterations) | Time (sec) |
|:---:|:---:|
| NOT | 64,763.4 |
| RCOT | 117,126.1 |
| OTUR | 24,255.6 |
| Ours (UOTIP) | 52,326.2 |

(a) Training time comparison

| Method | Parameters (M) |
|:---:|:---:|
| NOT | 65.934 |
| RCOT | 77.442 |
| OTUR | 20.369 |
| Ours (UOTIP) | 31.474 |

(b) Number of parameters.

Table 3: Comparison of training efficiency and model complexity across different methods.

## C  TWIST CONDITION

In this section, we discuss conditions ensuring the existence of the OT map. First, we introduce the condition referred to as the *twist condition*.

**Definition C.1** (Left twist condition). *Let $M$ and $N$ be a $n$-dimensional manifold and $N$ be a Polish space. Let $c : M \times N \to \mathbb{R}$ be a cost function and $\mu$ and $\nu$ be two probability measures on $M$ and $N$, respectively. For a given cost function $c(\mathbf{y}, \mathbf{x})$, we define the skew left Legendre transform as the partial map*

$$\Lambda_c^l : M \times N \to T^*M, \quad \Lambda_c^l(\mathbf{y}, \mathbf{x}) = \left(\mathbf{y}, \frac{\partial c}{\partial \mathbf{y}}(\mathbf{y}, \mathbf{x})\right)$$

*whose domain of definition is*

$$\mathcal{D}(\Lambda_c^l) = \left\{(\mathbf{y}, \mathbf{x}) \in M \times N : \frac{\partial c}{\partial \mathbf{y}}(\mathbf{y}, \mathbf{x}) \text{ exists}\right\}.$$

*We say that $c$ satisfies the left twist condition if $\Lambda_c^l$ is injective on $\mathcal{D}(\Lambda_c^l)$.*

Now, we state a theorem that guarantees the existence and uniqueness of the optimal transport map.

**Theorem C.1** ((Fathi & Figalli, 2010)). *Let $M$ be a smooth (second countable) manifold, and $N$ be a Polish space, and consider $\mu$ and $\nu$ (Borel) probability measures on $M$ and $N$ respectively. Assume that the cost $c : M \times N \to \mathbb{R}$ is lower semicontinuous and bounded from below. Moreover, assume that the following conditions hold:*

1. *the family of maps $\mathbf{y} \mapsto c(\mathbf{y}, \mathbf{x}) = c_{\mathbf{x}}(\mathbf{y})$ is locally semi-concave in $\mathbf{y}$ locally uniformly in $\mathbf{x}$,*

2. *the cost $c$ satisfies the left twist condition,*

3. *the measure $\mu$ gives zero mass to countably $(n-1)$-Lipschitz sets,*

4. *the infimum in the Kantorovitch problem $C(\mu, \nu) = \arg\min_{\gamma \in \prod}\{\int c(\mathbf{y}, \mathbf{x})d\gamma\}$ is finite.*

*Then there exists a borel map $T : M \to N$, which is an optimal transport map from $\mu$ to $\nu$ for the cost $c$. Moreover, the map $T$ is unique $\mu$-a.e., and any plan $\gamma_c \in \prod(\mu, \nu)$ optimal for the cost $c$ is concentrated on the graph of $T$.*

*Proof.* See (Fathi & Figalli, 2010). $\square$

**Remark 2.** *Denote $c(\mathbf{y}, \mathbf{x}; \lambda) = c_l(\mathbf{y}, \mathbf{x}) + \lambda c_q(\mathbf{y}, \mathbf{x})$ for $\lambda \geq 0$. Note that under our mild assumption in Section 2, all conditions of the above theorem except condition 2 (left twist condition) are satisfied with cost function $c(\mathbf{y}, \mathbf{x}; \lambda)$. To check the left twist condition, it is enough to show the injectivity of $\frac{\partial}{\partial \mathbf{y}}c(\mathbf{y}, \mathbf{x}; \lambda)$ with respect to $\mathbf{x}$. However, for $\lambda = 0$ (i.e., when only the likelihood-based cost term is used), the map $\mathbf{x} \mapsto \frac{\partial}{\partial \mathbf{y}}c(\mathbf{y}, \mathbf{x}; 0) = 2(\mathbf{y} - \mathcal{A}(\mathbf{x}))$ is not injective due to the ill-posedness of $\mathcal{A}$.*

*To handle this issue, we incorporate the standard quadratic cost $c_q$. Note that in many tasks such as Gaussian deblurring or HDR reconstruction, one can readily verify that $\frac{\partial}{\partial \mathbf{y}}c(\mathbf{y}, \mathbf{x}; \lambda) = 2(\mathbf{y} - \mathcal{A}(\mathbf{x})) + 2\lambda(\mathbf{y} - \mathbf{x})$ is injective with respect to $\mathbf{x}$ with $\lambda = 1$. In general, when $\mathcal{A}$ is Lipschitz continuous, we can choose $\lambda$ such that the function $\mathbf{x} \mapsto \frac{\partial}{\partial \mathbf{y}}c(\mathbf{y}, \mathbf{x}; \lambda)$ is injective.*

**Proposition C.1.** *Assume that $\mathcal{A}$ is $L$-Lipschitz continuous. Then the function $\mathbf{x} \mapsto \frac{\partial}{\partial \mathbf{y}}c(\mathbf{y}, \mathbf{x}; \lambda)$ is injective for all $\lambda > L$.*

*Proof.* Note that the following equation holds:

$$\frac{\partial}{\partial x}c(\mathbf{y}, \mathbf{x}; \lambda) = 2(\mathbf{y} - \mathcal{A}(\mathbf{x})) + 2\lambda(\mathbf{y} - \mathbf{x}) = (2 + 2\lambda)\mathbf{y} - 2(\lambda\mathbf{x} + \mathcal{A}(\mathbf{x})). \quad (23)$$

Thus it is enough to show that $\lambda\mathbf{x} + \mathcal{A}(\mathbf{x})$ is injective. Also, for any $\mathbf{x}_1, \mathbf{x}_2 \in \mathcal{X}$, $\lambda\mathbf{x}_1 + \mathcal{A}(\mathbf{x}_1) = \lambda\mathbf{x}_2 + \mathcal{A}(\mathbf{x}_2)$ implies that $\lambda\|\mathbf{x}_1 - \mathbf{x}_2\| = \|\mathcal{A}(\mathbf{x}_1) - \mathcal{A}(\mathbf{x}_2)\| \leq L\|\mathbf{x}_1 - \mathbf{x}_2\|$. Thus letting $\lambda > L$, the above result implies that $\mathbf{x}_1 = \mathbf{x}_2$ and $\lambda\mathbf{x} + \mathcal{A}(\mathbf{x})$ is injective. $\square$

## D RELATED WORK

Inverse problems have been investigated from diverse approaches, from traditional methods, such as hand-crafted priors and sparse coding, to deep learning techniques that learn complex image representations from large datasets.

Hand-crafted priors characterize natural images using predefined structures and simple statistical properties, such as sparsity (Candès & Wakin, 2008), low-rank representations (Fazel et al., 2008), or total variation (Candès et al., 2006). Although these methods do not require training data, they struggle to capture the complexity of natural image distributions and often fail to reliably distinguish realistic structures from unnatural artifacts. Sparse coding and dictionary learning partially mitigate this limitation by learning dictionaries from training data, but remain limited in capturing complex image structures (Qayyum et al., 2022).

With the success of deep learning, supervised approaches have achieved notable advances in many inverse problems, including image denoising (Zhang et al., 2017), super-resolution (Dong et al., 2014), and motion deblurring (Nah et al., 2017). These methods directly learn mappings from corrupted observations to clean ground-truth signals using paired datasets. However, these supervised methods rely on large collections of paired noisy/clean images, which are expensive or infeasible to obtain in many domains. Moreover, they often generalize poorly to out-of-distribution measurements (Recht et al., 2019).

To address these challenges, unsupervised and self-supervised approaches have been proposed that eliminate the need for paired data by leveraging measurement consistency—enforcing that the reconstructed image $\hat{x}$, when passed through the forward operator $A$, reproduces the observed measurements $y$ (Ulyanov et al., 2018). More recently, generative models have emerged as powerful priors, such as GANs (Bora et al., 2017), VAEs (Goh et al., 2019), and diffusion models (Song et al., 2021; Chung et al., 2022; Zhang et al., 2025). These models learn the distribution of natural images, enabling reconstruction by combining learned priors with measurement consistency. While these approaches achieve high perceptual quality, they often require expensive pretraining and remain sensitive to mismatches in noise assumptions.

Furthermore, the learned regularizer approaches Lunz et al. (2018); Goujon et al. (2023) share a superficial similarity with OT-map–based approaches (Korotin et al., 2023; Tang et al., 2024) regarding the training loss forms. However, the underlying formulations and optimization objectives differ fundamentally. Lunz et al. (2018) incorporate a WGAN regularizer into the measurement loss, requiring a 1-Lipschitz discriminator and leading to a min–max optimization over the generator and discriminator. Goujon et al. (2023) further adopt a proximal-gradient reconstruction scheme utilizing the learned regularizer. In contrast, the OT-map-based approaches ((Korotin et al., 2023; Tang et al., 2024)) derive the training objective based on the semi-dual OT formulation. This results in a similar loss function **without Lipschitz constraints on the potential function, and the training becomes a max–min problem** that replaces the measurement loss with a quadratic cost function. This difference leads to fundamentally different theoretical properties, including the existence and uniqueness of solutions and the interpretation of the learned potential (Fan et al., 2023) and different practical performance (Choi et al., 2024). Importantly, the OT potential is defined between the **measurement distribution and the signal distribution**, whereas the GAN discriminator operates between the **true signal distribution and the generated distribution**, making the two notions conceptually and mathematically different.

In this paper, we introduce an inverse problem solver based on Unbalanced Optimal Transport (UOT). Unlike many existing methods, our method requires neither paired training data nor pretraining on large-scale natural image datasets. By incorporating a likelihood-based cost, our approach directly aligns reconstruction with the MAP estimate. Theoretically, the UOT framework relaxes the strict marginal-matching constraint, providing robustness to outliers, adaptability to multi-level observation noise and class imbalance, and generalization across diverse noise conditions. These properties make our method particularly suited for unpaired inverse problems, where only independent sets of noisy and clean samples are available. Persiianov et al. (2024)

Within the OT literature, Persiianov et al. (2024) investigates inverse optimal transport for semi-supervised learning. While this work shows some similarity through the use of likelihood terms, the problem formulations and objectives are fundamentally different. Persiianov et al. (2024) focuses on

inverse optimal transport, whose goal is to recover the cost function that makes a given map into an optimal transport map. In contrast, our goal is to learn the inverse mapping under a fixed measurement model.

# E ADDITIONAL QUALITATIVE EXAMPLES

## E.1 COMPARISON OF INVERSE PROBLEM SOLVERS ON AFHQ

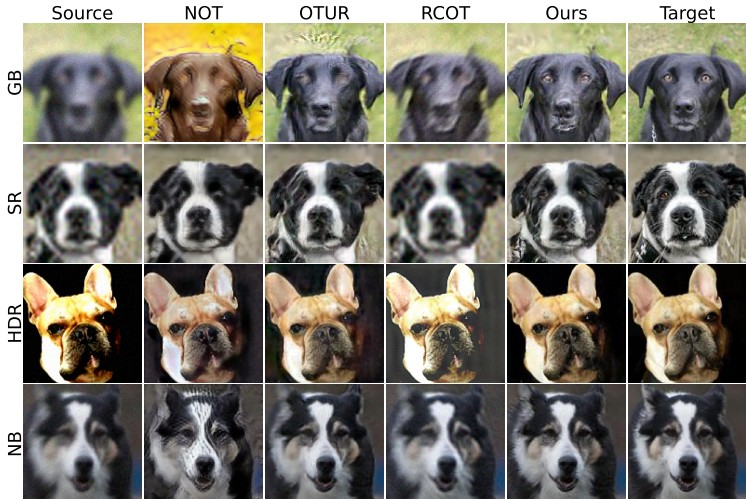

Figure 7: **Comparison of inverse problem solvers on AFHQ** for four tasks: Gaussian deblurring (GB), Super-resolution (SR), High dynamic range reconstruction (HDR), and Nonlinear deblurring (NB). Our model reconstructs images of superior fidelity with well-preserved structural details.

## E.2 RESULTS FOR VARIOUS INVERSE PROBLEMS

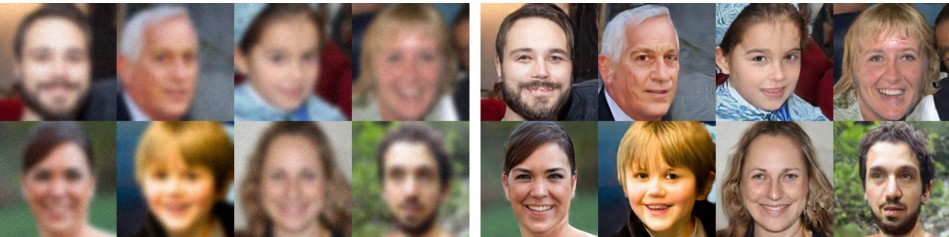

Figure 8: Additional qualitative results of our model for the Gaussian deblurring task on FFHQ (degraded (Left) → clean (Right)).

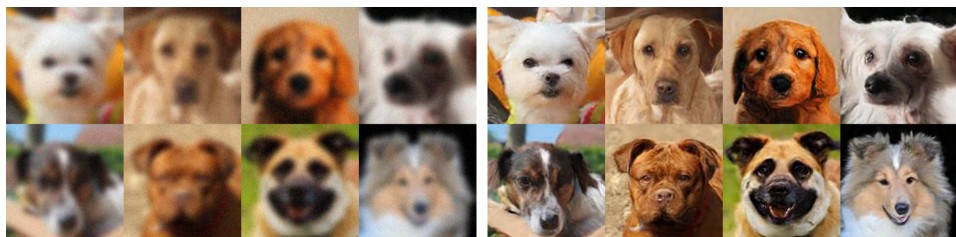

Figure 9: Additional qualitative results of our model for the Gaussian deblurring task on AFHQ (degraded (Left) → clean (Right)).

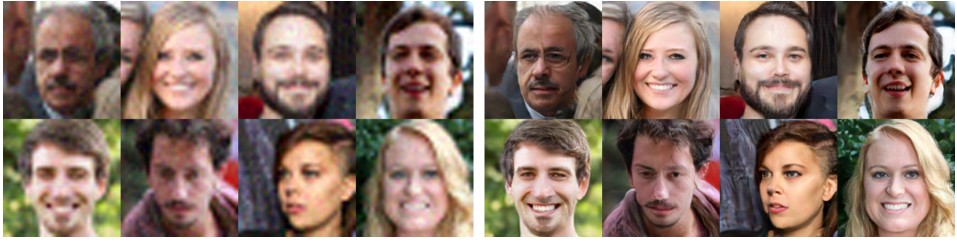

Figure 10: Additional qualitative results of our model for the super-resolution task on FFHQ (degraded (Left) → clean (Right)).

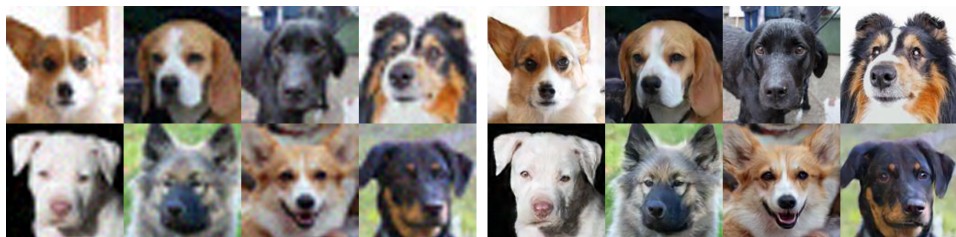

Figure 11: Additional qualitative results of our model for the super-resolution task on AFHQ (degraded (Left) → clean (Right)).

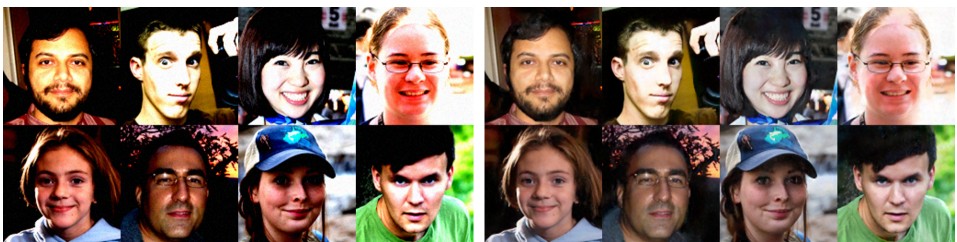

Figure 12: Additional qualitative results of our model for the HDR reconstruction task on FFHQ (degraded (Left) → clean (Right)).

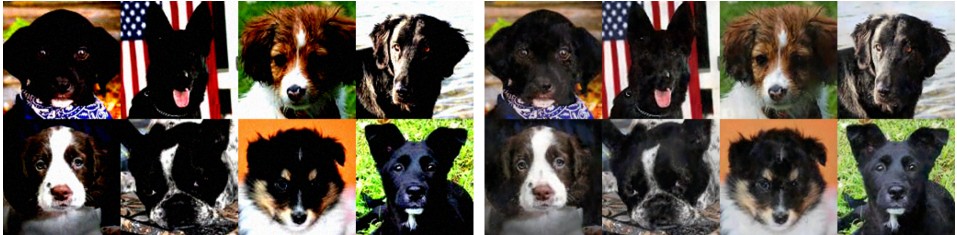

Figure 13: Additional qualitative results of our model for the HDR reconstruction task on AFHQ (degraded (Left) → clean (Right)).

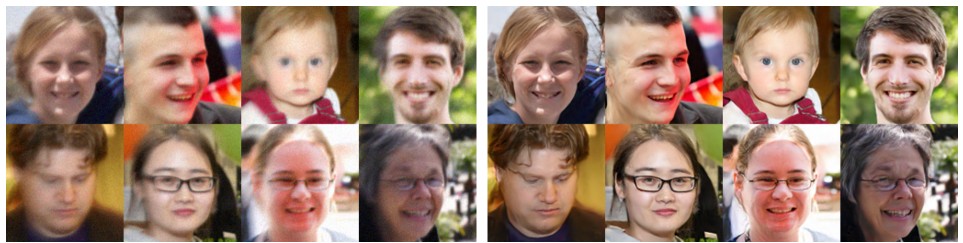

Figure 14: Additional qualitative results of our model for the nonlinear deblurring task on FFHQ (degraded (Left) → clean (Right)).

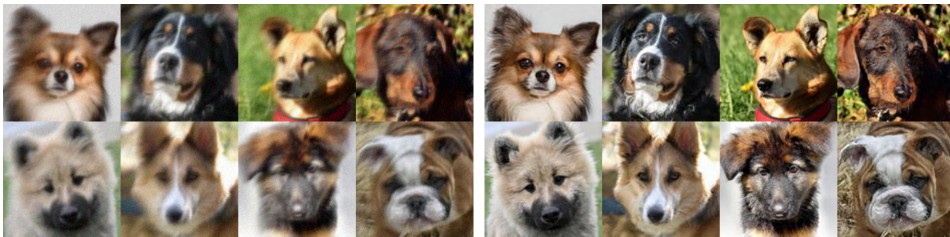

Figure 15: Additional qualitative results of our model for the nonlinear deblurring task on AFHQ (degraded (Left) → clean (Right)).

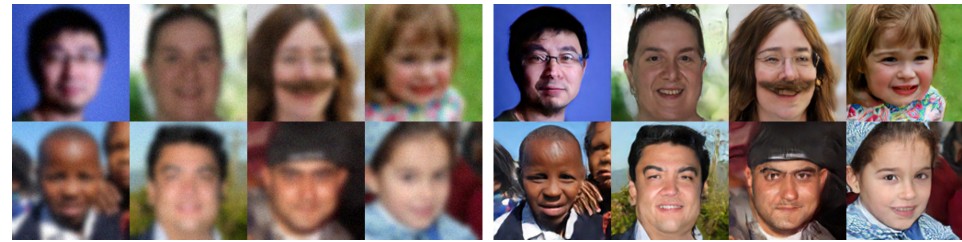

Figure 16: Qualitative results of our quadratic-cost-only variant for the Gaussian deblurring task on FFHQ (degraded (Left) → clean (Right)).

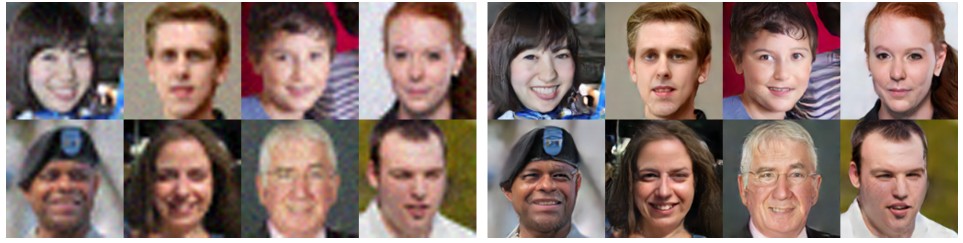

Figure 17: Qualitative results of our quadratic-cost-only variant for the super resolution task on FFHQ (degraded (Left) → clean (Right)).

### E.3 RESULTS FOR MULTI-LEVEL OBSERVATION NOISE

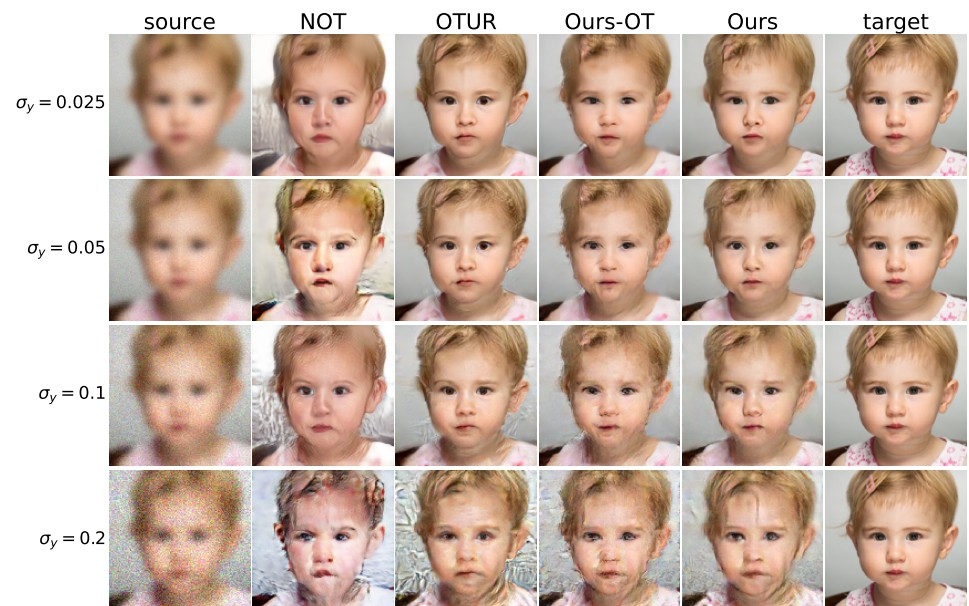

Figure 18: **Comparison of inverse problem solvers under multi-level observation noise** on FFHQ for the Gaussian deblurring.

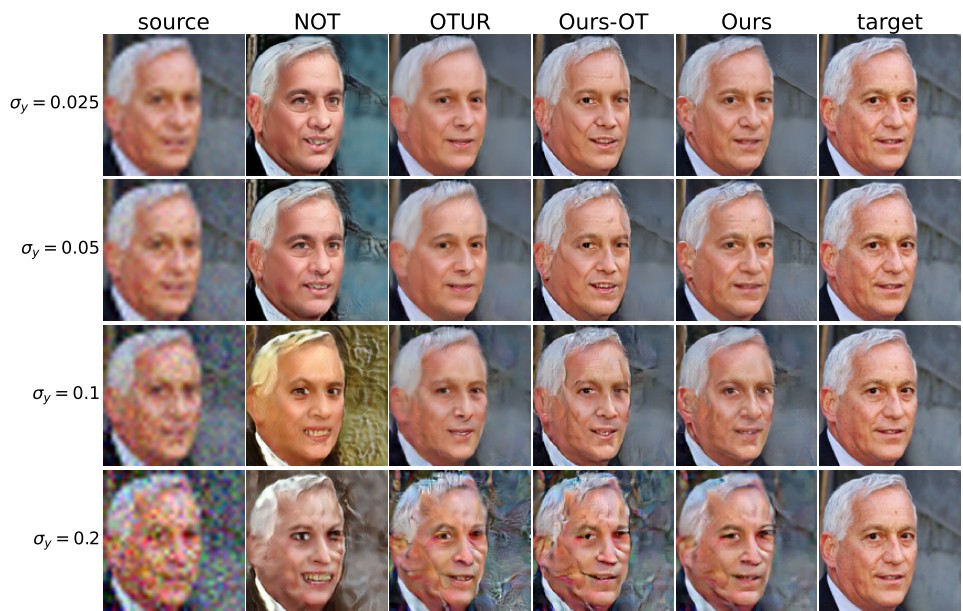

Figure 19: **Comparison of inverse problem solvers under multi-level observation noise** on FFHQ for the super-resolution.

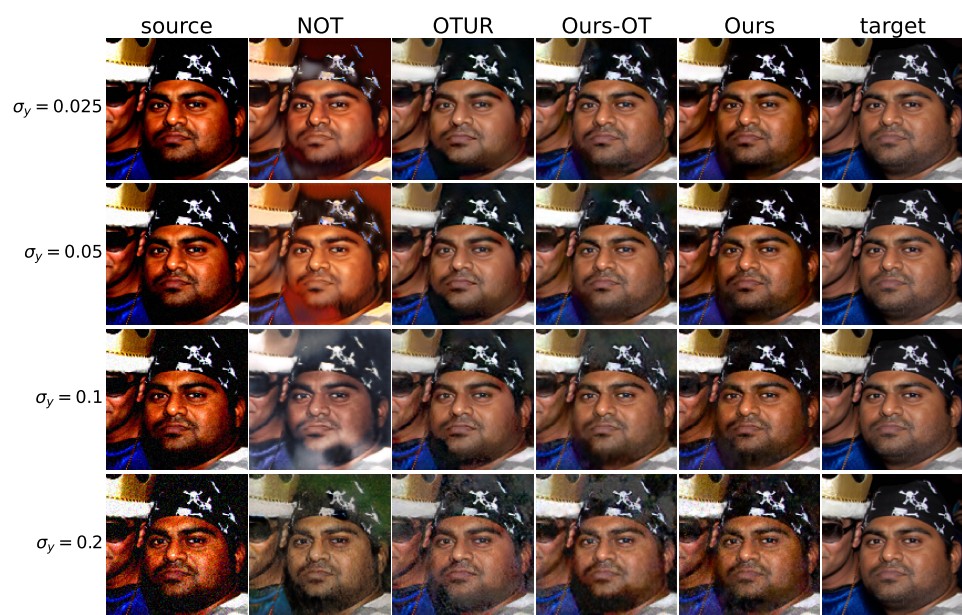

Figure 20: **Comparison of inverse problem solvers under multi-level observation noise** on FFHQ for the HDR reconstruction.

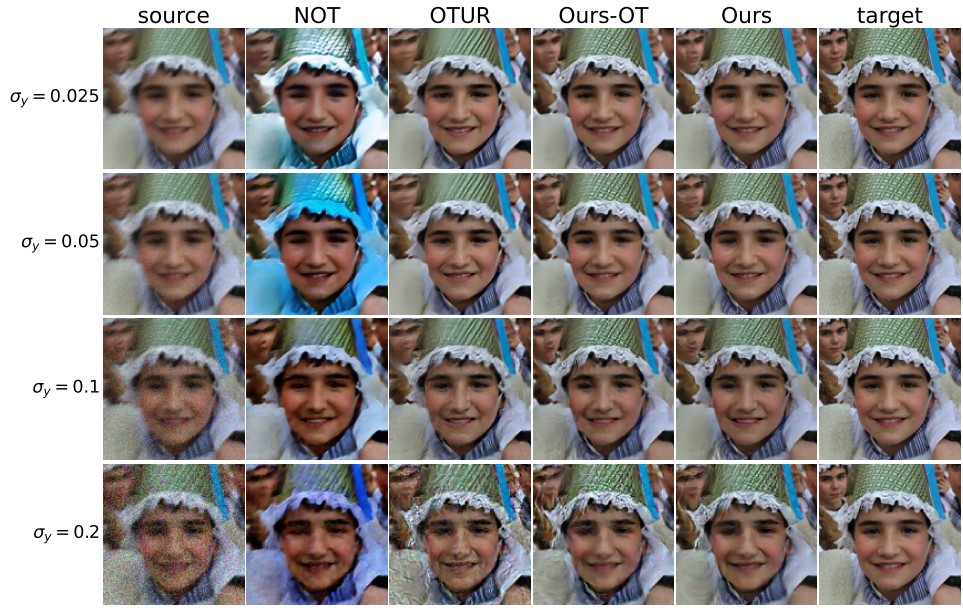

Figure 21: **Comparison of inverse problem solvers under multi-level observation noise** on FFHQ for the nonlinear deblurring.

# F ADDITIONAL QUANTITATIVE RESULTS

## F.1 RESULTS FOR COST INTENSITY PARAMETER ABLATION

We perform an ablation study on the cost intensity hyperparameter $\tau$ (Eq. 12) for two linear inverse problems on the FFHQ dataset (Fig. 22). We tested values $\tau \in 0.001 \times \{0.25, 0.5, 1, 2, 4\}$. Across both tasks, our model remains robust to the choice of $\tau$, except when $\tau$ is excessively large. Specifically, metrics that directly compare with ground-truth signals (PSNR, SSIM, and LPIPS) remain stable. On the other hand, the metric that measures marginal distribution-level fidelity (FID) degrades notably as $\tau$ increases. See Table 4 for the full result table.

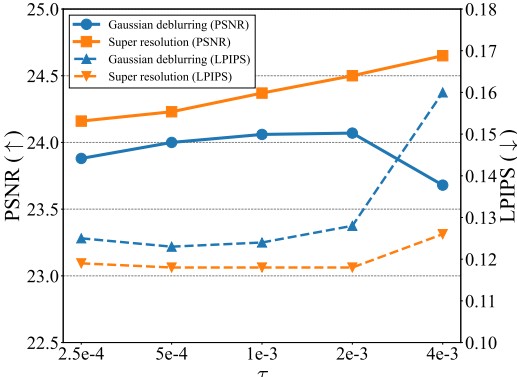

Figure 22: **Ablation study on the cost intensity parameter** $\tau$. For clarity, only PSNR ($\uparrow$) and LPIPS ($\downarrow$) are visualized.

Table 4: Ablation study on the cost intensity parameter $\tau$ with the FFHQ dataset. $\tau_0 = 1e - 3$.

| $\tau$ | Gaussian Deblurring | | | | Super Resolution | | | |
|---|---|---|---|---|---|---|---|---|
| | PSNR ($\uparrow$) | SSIM ($\uparrow$) | LPIPS ($\downarrow$) | FID ($\downarrow$) | PSNR ($\uparrow$) | SSIM ($\uparrow$) | LPIPS ($\downarrow$) | FID ($\downarrow$) |
| $0.25 \times \tau_0$ | 23.88 | 0.7104 | 0.125 | **18.580** | 24.16 | 0.7304 | 0.119 | **18.708** |
| $0.5 \times \tau_0$ | 24.00 | 0.7137 | **0.123** | 19.093 | 24.23 | 0.7338 | **0.118** | 18.824 |
| $\tau_0$ | 24.06 | **0.7139** | 0.124 | 21.210 | 24.37 | 0.7366 | **0.118** | 20.234 |
| $2 \times \tau_0$ | **24.07** | 0.7092 | 0.128 | 30.739 | 24.50 | 0.7405 | **0.118** | 21.277 |
| $4 \times \tau_0$ | 23.68 | 0.6815 | 0.160 | 74.994 | **24.65** | **0.7426** | 0.126 | 30.281 |

## F.2 QUANTITATIVE RESULT TABLES

Table 5: **Quantitative results under multi-level observation noise** for four inverse problems on FFHQ. Our model exhibits superior robustness across noise levels.

(a) Nonlinear inverse problems: High Dynamic Range and Nonlinear Deblurring.

| Method | High Dynamic Range | | | | Nonlinear Deblurring | | | |
|---|---|---|---|---|---|---|---|---|
| | PSNR (↑) | SSIM (↑) | LPIPS (↓) | FID (↓) | PSNR (↑) | SSIM (↑) | LPIPS (↓) | FID (↓) |
| NOT (Korotin et al., 2023) | 21.01 | 0.7728 | 0.153 | 62.641 | 21.50 | 0.7201 | 0.177 | 76.795 |
| OTUR (Wang et al., 2022) | 24.29 | 0.8098 | 0.103 | 52.149 | 25.23 | 0.7763 | 0.114 | 61.244 |
| OTIP (Ours-OT) | **25.92** | **0.8488** | 0.086 | **49.750** | 26.90 | 0.8307 | 0.087 | 51.805 |
| UOTIP (Ours) | 25.44 | 0.8330 | **0.084** | 51.300 | **27.25** | **0.8427** | **0.077** | **43.450** |

Table 6: **Class imbalance results under imbalance ratio** $k$ between AFHQ-cat and AFHQ-dog for the Gaussian deblurring.

| Ratio | Model | PSNR (↑) | SSIM (↑) | LPIPS (↓) | FID (↓) |
|---|---|---|---|---|---|
| 1 | NOT | 20.28 | 0.5567 | 0.248 | 50.166 |
| | OTUR | 23.13 | 0.6477 | 0.179 | 28.626 |
| | OTIP (Ours-OT) | 23.15 | 0.6373 | 0.165 | 28.182 |
| | UOTIP (Ours) | **23.42** | **0.6523** | **0.153** | **21.558** |
| 2 | NOT | 19.10 | 0.5073 | 0.285 | 60.739 |
| | OTUR | 22.92 | 0.6435 | 0.184 | 32.429 |
| | OTIP (Ours-OT) | 22.64 | 0.6190 | 0.180 | 34.047 |
| | UOTIP (Ours) | **23.29** | **0.6438** | **0.160** | **25.689** |
| 3 | NOT | 18.35 | 0.4977 | 0.311 | 66.060 |
| | OTUR | 22.19 | 0.6182 | 0.198 | 36.812 |
| | OTIP (Ours-OT) | 22.50 | 0.6128 | 0.195 | 66.842 |
| | UOTIP (Ours) | **23.04** | **0.6315** | **0.172** | **31.630** |
| 4 | NOT | 18.11 | 0.4674 | 0.320 | 58.237 |
| | OTUR | 20.92 | 0.5671 | 0.227 | 40.651 |
| | OTIP (Ours-OT) | 22.42 | 0.6045 | 0.208 | 78.869 |
| | UOTIP (Ours) | **22.67** | **0.6155** | **0.190** | **39.894** |

Table 7: **Quantitative results under diverse noise types** in linear inverse problems on the FFHQ dataset.

| Ratio | Method | Gaussian Deblur | | | | Super Resolution 4× | | | |
|---|---|---|---|---|---|---|---|---|---|
| | | PSNR (↑) | SSIM (↑) | LPIPS (↓) | FID (↓) | PSNR (↑) | SSIM (↑) | LPIPS (↓) | FID (↓) |
| Gaussian noise | NOT | 20.11 | 0.6035 | 0.209 | 52.901 | 20.13 | 0.6257 | 0.209 | 50.066 |
| | OTUR | 23.82 | 0.7106 | 0.136 | 24.337 | 24.09 | 0.7243 | 0.129 | 22.751 |
| | UOTIP (Ours) | **24.06** | **0.7139** | **0.124** | **21.210** | **24.35** | **0.7371** | **0.118** | **19.475** |
| Laplace noise | NOT | 19.50 | 0.5810 | 0.234 | 57.921 | 20.41 | 0.6277 | 0.208 | 50.228 |
| | OTUR | 23.63 | 0.7074 | 0.137 | 23.825 | 23.72 | 0.7252 | 0.130 | 23.931 |
| | UOTIP (Ours) | 23.98 | 0.7153 | 0.123 | **21.846** | **24.04** | **0.7350** | **0.118** | **19.629** |
| | UOTIP (Ours-Laplace's likelihood) | **24.12** | **0.7204** | **0.121** | 25.342 | 23.99 | 0.7317 | 0.122 | 27.053 |
| Poisson noise | NOT | 19.15 | 0.5584 | 0.246 | 55.368 | 19.28 | 0.5820 | 0.248 | 57.606 |
| | OTUR | 23.14 | 0.6919 | 0.149 | 25.840 | 23.26 | 0.7020 | 0.148 | 26.222 |
| | UOTIP (Ours) | **23.47** | **0.6938** | **0.136** | **23.669** | **23.59** | **0.7163** | **0.136** | **20.396** |
| | UOTIP (Ours-Poisson likelihood) | 18.06 | 0.4295 | 0.403 | 290.053 | 22.56 | 0.6075 | 0.351 | 175.036 |

Table 8: Ablation study on the cost function $c(\mathbf{y}, \mathbf{x})$, investigated on the FFHQ dataset

| | cost term | | Gaussian Deblurring | | | | Super Resolution 4× | | | |
|---|---|---|---|---|---|---|---|---|---|---|
| | Quad term | IP term | PSNR (↑) | SSIM (↑) | LPIPS (↓) | FID (↓) | PSNR (↑) | SSIM (↑) | LPIPS (↓) | FID (↓) |
| Ours | ✓ | | 24.01 | 0.7130 | 0.124 | 21.516 | 24.29 | 0.7332 | 0.119 | 20.131 |
| | | ✓ | 23.98 | **0.7140** | 0.127 | 25.608 | 24.22 | 0.7354 | **0.117** | **19.085** |
| | ✓ | ✓ | **24.06** | 0.7139 | **0.124** | **21.210** | **24.35** | **0.7371** | 0.118 | 19.475 |
| OTUR (Wang et al., 2022) | | | 23.82 | 0.7106 | 0.136 | 24.337 | 23.91 | 0.6777 | 0.165 | 30.773 |

