# OpenReview forum: "UOTIP: Unbalanced Optimal Transport Map for Unpaired Inverse Problems"
_ICLR.cc/2026/Conference — Submitted to ICLR 2026_

### Official Review · Reviewer_nmHQ · 2025-10-20

**Soundness:** 3
**Presentation:** 3
**Contribution:** 2
**Rating:** 2
**Confidence:** 4

**Summary:**

The paper presents a novel inverse problem solver based on unbalanced optimal transport (UOT), which can learn from unpaired datasets of measurements and ground-truth signals. The paper motivates a loss based on UOT that requires training two networks in a minimax way, the inverse solver $T$ and a potential function $v$. The paper compares with other inverse solvers based on optimal transport, showing competitive results on super-resolution, Gaussian deblurring, high-dynamic range reconstruction, and non-linear deblurring.

**Strengths:**

- Introduces an interesting loss for training reconstruction networks based on optimal transport that can handle class imbalances and unpaired data.
- Presents better reconstruction performances than other optimal-transport baselines in various inverse problems.
- A series of ablations of the proposed method are presented that illustrate the impact on unbalanced datasets and robustness across noise types and varying noise levels.

**Weaknesses:**

- A discussion of links to the state-of-the-art outside optimal transport is lacking: there exist various approaches that can train reconstruction networks with unpaired datasets, and it would add a significant amount of value to the paper to include a discussion on the links and differences with these methods, and also some numerical comparisons.
    - learned regularisers: There are various papers learning regularizers from unpaired data, e.g., "Adversarial regularizers in inverse problems", NeurIPS 2018, or "A neural-network-based convex regularizer for inverse problems." IEEE TCI 2023. it would be good to understand how the potential function used in this paper relates to them.
    - GAN-based: CycleGAN also doesn't require paired data that have a close relationship to optimal transport e.g. "Optimal Transport driven CycleGAN for Unsupervised Learning in Inverse Problems", SIAM 2019.
    - diffusion/PnP priors: there is an extensive literature on training denoising-based priors (aka Tweedie's formula), which also doesn't require paired data (one could add noise to the ground-truth to learn the denoiser).

- The method seems to fail in noisy settings $\sigma_y\geq 0.1$ (see e.g. fig. 17), while most state-of-the-art learned inverse problem solvers can handle this amount of noise.

- I have a few concerns regarding the blind inverse problem setting:
    - The size of the measurements often doesn't match the size of the signals/images. This should be clarified from the beginning and offer a generic solution. This is only mentioned for the SR experiments, and it is unclear how one would resize measurements in other blind scenarios.
   - The experiments for the blind case are only reported in tables. It would be good to show some visual results to understand how well the algorithm performs.

- The presentation of the experiments could be improved:
   - The Poisson noise definition is unclear, and the reader is pointed to (the appendix!) of another paper to understand it. After reading the cited paper, I understood that the rate is defined w.r.t. to 8-bit pixel values $[0,255]$. This differs from the Gaussian noise definition, which is on normalized values $[0,1]$. This is non-consistent and confusing. I recommend including a clear definition of the Poisson noise in the paper.
   - the non-linear operator is not described either - it feels that the paper 'copied-pasted' some degradation operators from other papers without making an effort to explain them to the reader.


- The figures/tables could be improved: Fig. 1 is a simple description of what is an inverse problem is, and does not convey any specific information about the paper. Fig. 5 would be clearer as a table instead of a bar plot.

- I believe the loss could be better explained to a wider machine learning audience by simplifying the notation in eq. 15: I would replace the values of $\Psi_i$ by one of the two options considered in the paper, replace integrals by sums over a finite dataset (or write expectations).

- The idea of class imbalances is not very common in inverse problems to the best of my knowledge. I would suggest to explain the motivation early on in the paper.

- I believe the unpaired setting with a known forward operator might not be sufficiently realistic: even if the noise level is unknown, one could generate paired data by applying the degradation operator at multiple noise levels to the ground-truth data, and then train an end-to-end network on this paired dataset. Note that it is a standard practice to train end-to-end networks on multiple noise levels.

**Questions:**

- It is unclear how the datasets of measurements and signals are created: is there overlap between the measurement and signal datasets? More specifically, can we find the noisy/degraded version of a clean signal in the measurement dataset? I would expect that a fair, unpaired dataset has no overlap, since otherwise one could try to pair them by 'manually' (or with a simple ad-hoc distance/metric), looking for the closest measurement to any given ground truth signal.

- The two conditions 3.1 seem to contradict the perception-distortion trade-off (Blau and Michaeli, 2019) that states that one can't recover the signal distribution exactly - condition i - and at the same time have low distortion - condition ii - (i.e., low error wrt ground-truth) reconstructions?

- The multi-level noise experiment doesn't seem very realistic: in such an experiment, why wouldn't one average the multiple noise realisations of the same image in the dataset? This would boost the SNR and remove the unbalanced issue.

- It is not fully clear to me whether the optimization wrt $T_\theta$ is done inside the integral of $y$ or outside in practice, i.e. do you minimize the objective for each sample $y$ or just on average over the $y$ distribution?

- The resizing operation in the blind SR problem doesn't break the twist condition?

- Would you still expect the algorithm to work in blind settings where the simple quadratic loss $\|x-y\|^2$ is not a good proxy for the error, i.e. motion deblurring, where $x$ can be shifted by one pixel to the right, leading to large quadratic errors?

---

> ### Author Response · Authors · 2025-11-21
> **Response to Reviewer nmHQ (1/6)**
>
> We sincerely thank the reviewer for reading our manuscript and providing valuable feedback. We hope our responses address the reviewer's concerns. Corresponding revisions are marked in Red.
>
> $ $
>
> ---
> > [W1] A discussion of links to the state-of-the-art outside optimal transport is lacking: there exist various approaches that can train reconstruction networks with unpaired datasets, and it would add a significant amount of value to the paper to include a discussion on the links and differences with these methods, and also some numerical comparisons.
>
> >> * learned regularisers: There are various papers learning regularizers from unpaired data, e.g., "Adversarial regularizers in inverse problems", NeurIPS 2018, or "A neural-network-based convex regularizer for inverse problems." IEEE TCI 2023. it would be good to understand how the potential function used in this paper relates to them.
> >> * GAN-based: CycleGAN also doesn't require paired data that have a close relationship to optimal transport e.g. "Optimal Transport driven CycleGAN for Unsupervised Learning in Inverse Problems", SIAM 2019
> >> * diffusion/PnP priors: there is an extensive literature on training denoising-based priors (aka Tweedie's formula), which also doesn't require paired data (one could add noise to the ground-truth to learn the denoiser).
>
> **A.** We appreciate the reviewer for suggesting the important related works. Following the reviewer's suggestion, we (1) conducted additional comparisons to a GAN-based baseline (OT-cycleGAN), (2) included the discussion on learned regularizer methods in the related works, and (3) clarified why diffusion/PnP priors belong to a fundamentally different category from UOTIP.
>
> ### (1) Additional experiments with a GAN-based unpaired method
> We evaluated OT-CycleGAN [1] on both a linear inverse problem (Gaussian deblurring) and a nonlinear inverse problem (HDR reconstruction). Results show that UOTIP achieves significantly stronger performance.
> > **Gaussian Deblur**
>
> |           | Method                      | PSNR ↑ | SSIM ↑  | LPIPS ↓ | FID ↓   |
> |:--|--|--|--|--|--|
> |    FFHQ | OT-cycleGAN                        | 16.34 | 0.4813 | 0.314 | 140.256 |
> |  | NOT                        | 20.11  | 0.6035 | 0.209 | 52.901 |
> |                | OTUR                        | _23.82_ | _0.7106_ | _0.136_ | _24.337_ |
> |                | **UOTIP (Ours)**          | **24.06** | **0.7139** | **0.124** | **21.210** |
>
> > **HDR Reconstruction**
>
> |           | Method                      | PSNR ↑ | SSIM ↑  | LPIPS ↓ | FID ↓   |
> |:--|--|--|--|--|--|
> |    FFHQ | OT-cycleGAN                        | 17.93 | 0.6711 | 0.231 | 87.355 |
> |  | NOT                        | 21.24  | 0.7978  | 0.138   | 25.842  |
> |                | OTUR                        | _25.32_ | _0.8545_ | _0.076_ | **16.458** |
> |                | **UOTIP (Ours)**          | **26.02** | **0.8642** | **0.064** | _20.840_ |
>
> ### (2) Discussion of learned-regularizer approaches (added in Related Work (Appendix D))
>
> We included a discussion clarifying the conceptual differences between learned-regularizer methods [2, 3] and OT-map methods [NOT, RCOT]. In summary, while both families may yield training losses with similar forms, their **underlying mathematical formulations and optimization objectives are fundamentally different**.
>
> Learned-regularizer methods rely on WGAN-style discriminators with 1-Lipschitz constraints and are trained via min–max optimization. OT-map approaches, in contrast, arise from the semi-dual optimal transport formulation, leading to a max–min objective without Lipschitz constraints. Consequently, the learned potential in OT does not share the same theoretical role as the GAN discriminator. The OT potential is defined between the measurement and signal distributions, whereas the GAN discriminator operates between the true and generated signal distributions.
>
> Thus, although the training procedures may appear similar, the two approaches differ both mathematically and conceptually.
>
> ### (3) Why UOTIP is different from diffusion / PnP prior methods
>
> We agree with the reviewer that diffusion/PnP prior approaches form another major category of unpaired solvers. However, these approaches follow a fundamentally different modeling paradigm with iterative sampling procedures, whereas UOTIP is a direct transport method (with NFE 1) grounded in Unbalanced Optimal Transport. Because the two paradigms differ fundamentally in inference mechanism, training objective, and computational cost, we consider them complementary rather than directly comparable.
>
> [1] Sim, Byeongsu, et al. "Optimal transport driven CycleGAN for unsupervised learning in inverse problems." SIAM Journal on Imaging Sciences 13.4 (2020): 2281-2306.
> [2] Lunz, Sebastian, Ozan Öktem, and Carola-Bibiane Schönlieb. "Adversarial regularizers in inverse problems." NIPS 2018.
> [3] Goujon, Alexis, et al. "A neural-network-based convex regularizer for inverse problems." IEEE TCI 2023.

---

> ### Author Response · Authors · 2025-11-21
> **Response to Reviewer nmHQ (2/6)**
>
> ---
> > [W2] The method seems to fail in noisy settings $\sigma_y \geq 0.1$ (see e.g. fig. 17), while most state-of-the-art learned inverse problem solvers can handle this amount of noise.
>
>  **A.** In Fig. 18 (same with Fig. 17 of initial submission), the experiment was conducted under a **multi-level noise setting**, where only a small proportion of samples is corrupted with high-intensity noise levels ($\sigma_y \geq  0.1$). The goal of this experiment is not to benchmark performance at a single noise level, but to evaluate whether the model can **adaptively handle heterogeneous noise**. This is a significantly more challenging scenario than uniform high noise. Under this setting, Fig. 18 shows that our model exhibits comparatively robust performance relative to other methods under the same conditions.
>
> $ $
>
> ---
> > [W3] I have a few concerns regarding the blind inverse problem setting:
> >>* The size of the measurements often doesn't match the size of the signals/images. This should be clarified from the beginning and offer a generic solution. This is only mentioned for the SR experiments, and it is unclear how one would resize measurements in other blind scenarios.
> >>* The experiments for the blind case are only reported in tables. It would be good to show some visual results to understand how well the algorithm performs.
>
> > [Q6] Would you still expect the algorithm to work in blind settings where the simple quadratic loss $||x-y||^2$ is not a good proxy for the error, i.e. motion deblurring, where can be shifted by one pixel to the right, leading to large quadratic errors?
>
> **A.** We would like to clarify that we do **not claim the quadratic-cost variant $c_{q}$ of UOTIP as a general blind inverse problem solver**. This variant was introduced purely as an ablation study on the choice of cost function and not as the main claim of the paper.
>
> Within this limited scope, the performance of $c_q$ demonstrates that a quadratic-cost formulation can be feasible in **blind settings when the measurement $y$ and signal $x$ share structural similarity**. This structural similarity makes the quadratic cost physically meaningful in the UOT formulation. Under this assumption on the forward operator $A$, a simple resizing operator such as bicubic interpolation is a reasonable and widely used way to match the dimensions of measurements and signals (as in the SR experiments).
>
> Moreover, following the reviewer's suggestion, we also included **qualitative results of blind case on FFHQ dataset** in the revised version of our manuscript (Fig. 16 (Gaussian deblurring) and Fig. 17 (Super resolution)). These visual results show that the quadratic-cost-only variant (for blind case) shows decent results, consistent with the quantitative metrics.
>
> Regarding Q6, in settings where the quadratic loss is not a good proxy for similarity (e.g., motion deblurring with spatial shifts), we would not expect the quadratic-cost-only variant to perform well, and we do not position it as suitable for such cases.

---

> ### Author Response · Authors · 2025-11-21
> **Response to Reviewer nmHQ (3/6)**
>
> ---
> > [W4] The presentation of the experiments could be improved:
> >> * The Poisson noise definition is unclear, and the reader is pointed to (the appendix!) of another paper to understand it. After reading the cited paper, I understood that the rate is defined w.r.t. to 8-bit pixel values [0, 255]. This differs from the Gaussian noise definition, which is on normalized values [0, 1]. This is non-consistent and confusing. I recommend including a clear definition of the Poisson noise in the paper.
> >> * the non-linear operator is not described either - it feels that the paper 'copied-pasted' some degradation operators from other papers without making an effort to explain them to the reader.
>
> **A.** Thank you for the careful comment regarding the presentation. Following the reviewer's suggestion, we added explicit definitions of both the Poisson noise model and the nonlinear operator in Appendix B as follows:
> 1. For Poisson noise, we now provide the full likelihood and clarify the pixel-domain conversion. This removes the need to consult external appendices and ensures consistency with the Gaussian-noise setup.
> > **Poisson noise** (Poisson noise level $\lambda = 1.0$):
> $$p( \mathbf{y} | \mathbf{x_0} ) = \prod_{j=1}^{n} \frac{{[A(\mathbf{x_0})]}_j^{y_j}  \exp(-[A(\mathbf{x_0})]_j) }{y_j !},$$
> > where $j \in [1, C \times H \times W]$. To be more specific, the Poisson noise is defined on integer pixel values $[0, 255]$. Thus, each normalized image (range $[-1,1]$) is converted to 8-bit $[0,255]$, Poisson noise is applied, and the result is rescaled back to $[-1,1]$ to form the measurements.
>
> 2. For the nonlinear operator, we clarified the formulation and data source as follows:
> > **Nonlinear deblurring** ($\mathcal{A}$: neural operator from (Tran et al., 2021)):
> $$
> \mathbf{y}= \mathcal{A}(\mathbf{x}) + \mathbf{n}, \; \mathbf{n} \sim \mathcal{N}(\mathbf{0}, \sigma_y^2 \mathbf{I}_{m}) \; \text{ for pretrained operator } \mathcal{A}
> $$
> Here, $\mathcal{A}$ is a pretrained neural operator model on the GoPro dataset [1], which learns to approximate the nonlinear blur characteristics observed in the dataset [2].
>
> [1] Nah, Seungjun, Tae Hyun Kim, and Kyoung Mu Lee. "Deep multi-scale convolutional neural network for dynamic scene deblurring." Proceedings of the IEEE conference on computer vision and pattern recognition. 2017.
> [2] Tran, Phong, et al. "Explore image deblurring via encoded blur kernel space." Proceedings of the IEEE/CVF conference on computer vision and pattern recognition. 2021.
>
> $ $
>
> ---
> > [W5] The figures/tables could be improved: Fig. 1 is a simple description of what is an inverse problem is, and does not convey any specific information about the paper. Fig. 5 would be clearer as a table instead of a bar plot.
>
> **A.** Following the reviewer's suggestion, we removed Fig. 1 to make room for incorporating the revisions into the main text.
>
> Regarding Fig. 5, we intentionally kept the bar-plot format because the relative comparison is crucial for illustrating (a) robustness under noise-type mismatch and (b) the cost-function ablation. To support clarity and completeness, we also provided the **full numerical results in Tables 7 and 8** in the appendix (Lines 460 and 468).
>
> $ $
>
> ---
> > [W6] I believe the loss could be better explained to a wider machine learning audience by simplifying the notation in eq. 15: I would replace the values of $\Psi_i$ by one of the two options considered in the paper, replace integrals by sums over a finite dataset (or write expectations).
>
> **A.** To improve algorithmic clarity, we included **Algorithm 1** in Appendix B (Lines 742-788), which presents the training loss using finite-sample expectations and avoids heavy notation. Also, right after Eq. 15, we now reference this implementation detail with:
> > For the training algorithm, refer to Algorithm 1.
>
> Regarding the notation of $\Psi_i$, these functions define the $f$-divergences $D_{\Psi_i}$ employed in the Unbalanced OT to penalize errors between the marginals of the transport plan and the source/target distributions (Eq. 6). Since our framework supports multiple choices of $f$-divergences, we chose to keep the general notation $\Psi_i$ in the main text and elaborate on the specific choices in Lines 267-269.

---

> ### Author Response · Authors · 2025-11-21
> **Response to Reviewer nmHQ (4/6)**
>
> ---
> > [W7] The idea of class imbalances is not very common in inverse problems to the best of my knowledge. I would suggest to explain the motivation early on in the paper.
>
> **A.** Thank you for the helpful suggestion. We revised the first part of class imbalance paragraph (Line 424) to better motivate its relevance in unpaired inverse problems as follows:
> > We further evaluate UOTIP under class imbalance, where source and target distributions are multi-modal but differ in their mode proportions. This situation can naturally occur in unpaired settings when measurements and clean signals are obtained from different data sources. For example, in super-resolution, large amounts of low-resolution data may be available, whereas high-resolution data may be limited. In this case, additional public high-resolution datasets may then be incorporated, leading class imbalance between modes. To examine this, we constructed datasets by combining AFHQ-cat with AFHQ-dog ...
>
> $ $
>
> ---
> > [W8] I believe the unpaired setting with a known forward operator might not be sufficiently realistic: even if the noise level is unknown, one could generate paired data by applying the degradation operator at multiple noise levels to the ground-truth data, and then train an end-to-end network on this paired dataset. Note that it is a standard practice to train end-to-end networks on multiple noise levels.
>
> **A.** We appreciate the reviewer for the insightful comment. We agree with the reviewer that, when the forward operator and noise type are fully known, we can generate **synthetic paired data** by applying $A$ to clean images and adding noise. To examine this scenario, we conducted additional experiments by augmenting our model with a supervised loss term:
> $$\mathcal{L} = \mathcal{L}_{unpaired} + \mathcal{L} _{sup} \quad \text{where} \quad \mathcal{L} _{sup} = \lVert T _{\theta}(y) - x \rVert _{2}^2$$
>
> Across both Gaussian deblurring and 4× super-resolution, UOTIP consistently improves when supervised pairs are added. These results show that **UOTIP integrates supervised information smoothly and benefits from paired augmentation when available**. The results are as follows:
>
> **Gaussian Deblur**
>
> |           | Method                      | PSNR ↑ | SSIM ↑  | LPIPS ↓ | FID ↓   |
> |:--|--|--|--|--|--|
> | FFHQ | NOT                        | 20.11  | 0.6035  | 0.209   | 52.901  |
> |                | OTUR                        | 23.82 | 0.7106 | 0.136 | 24.337 |
> |                | UOTIP (Ours)           | _24.06_ | _0.7139_ | _0.124_ | _21.210_ |
> |                | **UOTIP (Ours +paired)**        | **25.03**| **0.7574**     | **0.103** | **15.829**      |
>
> **Super Resolution 4x**
>
> |           | Method                      | PSNR ↑ | SSIM ↑  | LPIPS ↓ | FID ↓   |
> |:--|--|--|--|--|--|
> | **FFHQ** | NOT                        | 20.13  | 0.6257  | 0.209   | 50.066  |
> |                | OTUR                        | 24.09 | 0.7243 | 0.129 | 22.751 |
> |                | UOTIP (Ours)           | _24.35_ | _0.7371_ | _0.118_ | _19.475_ |
> |                | **UOTIP (Ours +paired)**        | **25.01**      | 	**0.7641**       | **0.102**       | **12.145**       |
>
> $ $
>
> However, **this strategy is not always feasible in practice**. The reviewer’s suggestion assumes that the noise distribution is known and can be simulated. In many real-world inverse problems, this is not the case. For example, in **compressed sensing MRI**, measurements follow:
> $$y_j = PFS_j x + n_j \in \mathbb{C}^n \text{ for }   j = 1, ..., J$$
> where for the $j$-th coil, each term is defined as follows:
> - $P \in$ {0,1}$^{m \times n}$ is a subsampling operator
> - $F$ is the Fourier transform
> - $S_j$ is the sensitivity map
> - $y_j$ is the measurement
> - $n_j$ is the noise
>
> In practice, MRI noise arises from multiple physical processes, including coil-related thermal noise, magnetic field inhomogeneity, eddy currents, signal decay, hardware delays [1, 2]. These effects produce structured, non-ideal, and difficult-to-model noise, and **cannot be faithfully simulated** using clean data [3]. Thus, generating reliable synthetic pairs is often infeasible.
>
> In contrast, **UOTIP is applicable**, because it does not require explicit knowledge of the noise distribution. It only requires the forward operator $A$ and has demonstrated robustness under noise-type mismatch and multi-level noise.
>
> [1] Lustig, M., Donoho, D., & Pauly, J. M. (2007). Sparse MRI: The application of compressed sensing for rapid MR imaging. Magnetic Resonance in Medicine: An Official Journal of the International Society for Magnetic Resonance in Medicine, 58(6), 1182-1195.
> [2] Uecker, Martin, et al. "ESPIRiT—an eigenvalue approach to autocalibrating parallel MRI: where SENSE meets GRAPPA." Magnetic resonance in medicine 71.3 (2014): 990-1001.
> [3] Fu, Zhiyang, et al. "A multi-scale residual network for accelerated radial MR parameter mapping." Magnetic resonance imaging 73 (2020): 152-162.

---

> ### Author Response · Authors · 2025-11-21
> **Response to Reviewer nmHQ (5/6)**
>
> ---
> >[Q1] It is unclear how the datasets of measurements and signals are created: is there overlap between the measurement and signal datasets? More specifically, can we find the noisy/degraded version of a clean signal in the measurement dataset? I would expect that a fair, unpaired dataset has no overlap, since otherwise one could try to pair them by 'manually' (or with a simple ad-hoc distance/metric), looking for the closest measurement to any given ground truth signal.
>
> **A.** Thank you for the helpful question. In our experiments, the measurement dataset is created by applying the forward operator $A$ to the clean-signal dataset. This necessarily creates overlap in the underlying sources. However, the setting is still unpaired. **No pairing information is provided to the model**. During training, measurements $\{y\}$ and clean signals $\{x\}$ are treated as independent datasets, i.e., the mini-batches are sampled independently. We do not retain or use index-level correspondence, and the algorithm has no access to any matched pairs. We added clarification in the revised manuscript to make this explicit as follows in Appendix B:
> > For FFHQ, we use 6,000 images from the original dataset as the clean signal, allocating 5,500 to the training set and 500 to the test set. For AFHQ, we use the original training set and the original test set as the clean signal. The measurements are generated by applying the forward operator (Eq. (1)) to these clean signals. Note that under the unpaired assumption, mini-batches of measurement $Y$ and clean signal $X$ are always independently sampled in Algorithm 1. For evaluation, PSNR, SSIM, and LPIPS are computed on the test set, whereas FID is evaluated using both the training and test sets.
>
> $ $
>
> ---
> >[Q2] The two conditions 3.1 seem to contradict the perception-distortion trade-off (Blau and Michaeli, 2019) that states that one can't recover the signal distribution exactly - condition i - and at the same time have low distortion - condition ii - (i.e., low error wrt ground-truth) reconstructions?
>
> **A.** We appreciate the reviewer for the insightful question. We would like to clarify why Conditions (i) and (ii) in Sec. 3.1 do not contradict the perception–distortion trade-off of Blau & Michaeli (2019). The key points are: (1) **our two conditions have a strict hierarchy** and (2) **condition (ii) does not correspond to distortion**.
>
> (1) As a reminder, Monge's OT problem is defined as follows:
> $$C(\mu, \nu) := \inf_{T_\sharp \mu = \nu}  \left[ \int_{\mathcal{Y} } c(y,T(y)) d \mu (y) \right]$$
>
> The optimal transport map $T^{\star}$ is defined as the above transport cost minimizer. Therefore, the condition (i) (i.e., $T_{\sharp} \mu = \nu$) is a hard constraint for the transport map. The condition (ii) (i.e., transport cost minimization) is applied within the set of transport maps already satisfying (i). In other words, the condition (ii) does not attempt to reduce transport cost independently. It selects the best transport map among those that already satisfy the strict distribution-matching constraint (i). **Because the perception–distortion trade-off arises when perception and distortion are optimized simultaneously, this hierarchical structure of OT avoids the trade-off phenomenon**.
>
> (2) Moreover, the condition (ii) does not correspond to the distortion (i.e., error wrt ground-truth). Our likelihood cost $c_{l}(\cdot, \cdot)$ is defined as follows:
> $$c_{l}(y, T_\theta(y)) = \lVert \mathcal{A}(T_{\theta}(y)) - y\rVert_2^2$$
>
> This cost measures the error between the given measurement $y$ and the forward operator output of the prediction $x_{pred} = T_{\theta}(y)$. In contrast, the distortion indicates the error between $x_{gt}$ and $x_{pred}$, i.e., $d (x_{gt}, x_{pred})$. Therefore, **our likelihood cost minimization is not equivalent to the distortion minimization**.

---

> ### Author Response · Authors · 2025-11-21
> **Response to Reviewer nmHQ (6/6)**
>
> ---
> >[Q3] The multi-level noise experiment doesn't seem very realistic: in such an experiment, why wouldn't one average the multiple noise realisations of the same image in the dataset? This would boost the SNR and remove the unbalanced issue.
>
> **A.** Our multi-level noise experiment is designed to model scenarios where each measurement is acquired under a different noise condition, which is common in real systems. In many imaging systems, **the noise level varies across measurements** due to factors such as differences in measurement devices, exposure settings, or sensor temperature. As a result, each $A(x)$ is naturally observed under a distinct noise level, making this heterogeneous-noise setup a realistic and practically relevant evaluation.
>
> Regarding the reviewer’s suggestion of averaging multiple noisy realizations of the same clean signal $x$, this strategy can create **distribution shift challenge** in practice. Formally, if multiple measurements $y_{j}(x_{i})$ exist for each clean signal $x_i$, averaging them produces a new measurement $\tilde{y}(x_i)$:
> $$
> \tilde{y}(x_i) = \frac{1}{J} \sum_j  y_j(x_i),
> $$
> But, this $\tilde{y_{i}}$ follows a noise distribution that is **different from any of the original $y_{j}(x_{i})$**. Moreover, **this noise distribution will vary depending on each sample $x_{i}$**. Therefore, at test time, the model must generalize to noise levels that it has never observed during training, creating a distribution shift.
>
> For these reasons, we believe that evaluating robustness under the multi-level noise is a realistic and practically important setting.
>
> $ $
>
> ---
> > [Q4] It is not fully clear to me whether the optimization wrt $T_{\theta}$ is done inside the integral of $y$ or outside in practice, i.e. do you minimize the objective for each sample $y$ or just on average over the $y$ distribution?
>
>  **A.** As discussed in the response to W6, to improve algorithmic clarity, we included **Algorithm 1** in Appendix B. The optimization with respect to $T_{\theta}$ is performed over the mini-batch average of sampled measurements $y$.  In practice, this is equivalent to minimizing the objective for each sample $y$, because the population minimizer $T^{\star}$ is defined as the pointwise minimizer for each individual $y$.
>
> $ $
>
> ---
> > [Q5] The resizing operation in the blind SR problem doesn't break the twist condition?
>
> **A.** Thank you for the careful question. In the super-resolution setting, we modify the quadratic cost by applying bicubic interpolation to match the low-resolution and high-resolution images (Lines 348). This interpolation operator causes the modified cost function to violate the twist condition. We clarified in the revised manuscript that this setting lies outside the theoretical assumptions, and should therefore be interpreted as a practical extension rather than a theoretically guaranteed case as follows (Page 7 footnote):
> > Note that under this modified cost, the twist condition in Remark 1 is no longer satisfied. Nevertheless, as shown in Table 1, UOTIP performs well on super-resolution in practice. We attribute this empirical success to the local smoothing inductive bias of the generator network architectures.

---

### Official Review · Reviewer_8dhT · 2025-10-23

**Soundness:** 3
**Presentation:** 3
**Contribution:** 3
**Rating:** 6
**Confidence:** 3

**Summary:**

The paper proposes an unbalanced optimal transport map for solving inverse problems, which targets unpaired imaging inverse problems with noisy measurements and clean images are sampled independently. Traditional OT enforces strict marginal matching between two distributions, however, the proposed method relaxes this via unbalanced OT, where mass is rescaled using divergence penalties. This allows to address challenges like imbalance datasets and noise level variations.

The proposed method combines a likelihood-based cost, enforcing measurement consistency, with a quadratic regularization term that satisfies the theoretical twist condition for the existence and uniqueness of the transport map.

The authors validate their approach on four inverse problems—Gaussian deblurring, super-resolution, HDR reconstruction, and nonlinear deblurring—showing that UOTIP consistently outperforms strong baselines such as NOT, OTUR, and RCOT. The paper’s main contributions include: (1) introducing UOT into the unpaired inverse problem setting, (2) formulating a principled likelihood-based transport cost grounded in inverse problem modeling,  and (3) demonstrating improved results with enhanced robustness to diverse noise and imbalance conditions.

**Strengths:**

* The paper is generally well-written and well-structured, with clear motivation and comprehensive ablation studies.
* The paper presents convincing and strong experimental evidence to verify the  claims of superiority of the performance of the proposed method.
* The provided experimental results are broad and inclusive (2 varied datasets, 4 inverse problems, and comparison with three relevant baselines).
* The performance of the proposed methods shows clear improvements over the existing baselines.
* The authors have included additional results demonstrating the ability of the proposed method to handle different noise types, noise levels and source and target data imbalance.
* The authors have provided ablation studies to assess the effect of cost functions, one problem adaptive which uses the measurement model and the second general quadratic cost function.

**Weaknesses:**

The core elements of the proposed framework—optimal transport maps, unbalanced OT, and likelihood-based cost formulations—already exist in the literature. The novelty of the paper therefore lies primarily in connecting and integrating these existing components within the context of unpaired inverse problems, rather than introducing a fundamentally new framework.

* The concept of using optimal transport maps (OT) for unpaired image tasks is not new. For example, the paper "An Optimal Transport Perspective on Unpaired Image Super‑Resolution (2022)" used the OT framework for unpaired SR. Thus, the baseline idea of “map distribution of measurement  to distribution of clean signal via OT” has been investigated.
* The concept of unbalanced optimal transport (UOT) itself is well‐established in the OT literature (e.g., handles mismatched masses, outliers) and has been applied in other ML contexts.
* Using likelihood‐based costs  appears in some related works (Inverse Entropic OT Solves Semi-supervised Learning via Data Likelihood Maximization (2024)).

* The paper  does not analyze training cost or scalability, which may become a concern for higher-resolution data.

**Questions:**

The proposed likelihood cost function  inherently depends on the properties of the forward operator A. Could the authors clarify what specific assumptions are required on A for the theoretical results to hold? Could the authors also comment on the cases where assumptions are not met in practice, but the method seems to work?
 In particular, does the framework extend to nonlinear or non-differentiable operators, or is it restricted to Lipschitz-continuous and differentiable forward models? How would the method behave for inverse problems where A is discontinuous, piecewise-defined, or only approximately known?
I believe a broader discussion of the characteristics of the inverse problems that could be addressed with the proposed method needs to be included in the paper.

---

> ### Author Response · Authors · 2025-11-21
> **Response to Reviewer 8dhT (1/2)**
>
> We sincerely thank the reviewer for carefully reading our manuscript and providing valuable feedback. We hope our responses to be helpful in addressing the reviewer's concerns. We highlighted the corresponding revisions in the manuscript in Yellow.
>
> $ $
>
> ---
> > The core elements of the proposed framework—optimal transport maps, unbalanced OT, and likelihood-based cost formulations—already exist in the literature. The novelty of the paper therefore lies primarily in connecting and integrating these existing components within the context of unpaired inverse problems, rather than introducing a fundamentally new framework.
>
> > [W1] The concept of using optimal transport maps (OT) for unpaired image tasks is not new. For example, the paper "An Optimal Transport Perspective on Unpaired Image Super‑Resolution (2022)" used the OT framework for unpaired SR. Thus, the baseline idea of “map distribution of measurement to distribution of clean signal via OT” has been investigated.
>
> > [W2] The concept of unbalanced optimal transport (UOT) itself is well‐established in the OT literature (e.g., handles mismatched masses, outliers) and has been applied in other ML contexts.
>
> > [W3] Using likelihood‐based costs appears in some related works (Inverse Entropic OT Solves Semi-supervised Learning via Data Likelihood Maximization (2024)).
>
> **A.** We thank the reviewer for pointing out the relevant prior work. We agree that OT maps, UOT formulations, and likelihood-based costs each have precedents in the literature. However, **none of these components has been integrated or analyzed within the unpaired inverse problem setting**. Our contributions lie precisely in building a principled and theoretically grounded framework for this domain. Specifically:
>
> 1. (W1) Prior OT-based works do not apply to general inverse problems. Prior OT approaches for unpaired image tasks (such as unpaired SR) were applied **directly between image distributions using a standard quadratic cost** [1]. These methods cannot address more complex nonlinear inverse problems, such as HDR and Nonlinear Deblurring, because they lack a **measurement-model–aware cost** that incorporates the forward operator $A$. Our framework introduces a **likelihood-based OT cost** that leverages the measurement model and noise type. Particularly, we analyze the existence issue of OT maps that arises from the ill-posedness of inverse problems and show that adding a simple quadratic regularization term resolves this issue (Remark 1). This is an important issue not addressed in prior work.
>
> 2. (W2) Prior UOT works do not address inverse problems. Although UOT is well-established, it has not been explored in the inverse-problem setting. Our method, UOTIP, is the first to **combine the informative likelihood cost with a regularizing quadratic term** (Remark 1) in order to both (i) guide the transport using measurement-model information and (ii) recover theoretical guarantees on map existence. This is fundamentally different from prior UOT-based generative modeling [2], where the quadratic cost has no practical meaning, or UOT-based point cloud completion [3], where an informative cost (infoCD) is used but without theoretical guarantees for the existence of a transport map.
> Moreover, while classical UOT theory discusses robustness under class imbalance, how this property manifests in inverse problems has never been studied. We reinterpret class-imbalance robustness in the inverse-problem context as robustness to **multi-level noise, which naturally occurs when measurements come from multiple sensors**. Our experiments verify that our model shows this robustness in practice.
>
> 3. (W3) Likelihood-based OT literature addresses a different problem. The suggested likelihood-based OT work focuses on inverse optimal transport, whose goal is to recover the cost function that makes a given map into optimal transport map [4]. In contrast, our goal is to learn the inverse mapping under a fixed measurement model. Although both involve likelihood terms, **the problem formulations and objectives are fundamentally different**. However, we agree with the reviewer that there are some similarities in terms of likelihood maximization. We added this work to the revised related-work section for clarity.
>
> In summary, although each component exists individually, **UOTIP is the first principled and theoretically justified integration of these ideas for unpaired inverse problems**, offering new guarantees and properties unavailable in prior work.
>
> [1] "An Optimal Transport Perspective on Unpaired Image Super‑Resolution (2022)"
> [2]  Choi, Jaemoo, Jaewoong Choi, and Myungjoo Kang. "Generative modeling through the semi-dual formulation of unbalanced optimal transport." NeurIPS 2023.
> [3] Lee, Taekyung, et al. "Unpaired Point Cloud Completion via Unbalanced Optimal Transport." ICML 2025.
> [4] Inverse Entropic OT Solves Semi-supervised Learning via Data Likelihood Maximization (2024)

---

> ### Author Response · Authors · 2025-11-21
> **Response to Reviewer 8dhT (2/2)**
>
> ---
> > [W4] The paper does not analyze training cost or scalability, which may become a concern for higher-resolution data.
>
>
> **A.** We appreciate the reviewer for the thoughtful comment. Following the reviewer's suggestion, we additionally included a comparison of **training time** and **parameter count** (as a proxy for memory consumption) in the Efficiency Analysis paragraph of the revised manuscript. OTUR is the most efficient baseline, and our UOTIP is the second most efficient. These results verify that all experiments were conducted under **equivalent and fair conditions**. The detailed results are as follows:
>
> |Training (60000 iteration)| Time (sec)|
> |:--- |:---|
> |NOT| 64,763.4|
> |RCOT| 117,126.1|
> |OTUR| 24,255.6 |
> |Ours (UOTIP)| 52,326.2 |
>
> $ $
>
> ---
> > [Q1] The proposed likelihood cost function inherently depends on the properties of the forward operator $A$. Could the authors clarify what specific assumptions are required on A for the theoretical results to hold?
>
> **A.** Thank you for the careful question. As discussed in Remark 1, our theoretical results require the forward operator $A$ to be **Lipschitz continuous and differentiable**. Lipschitz continuity is required to guarantee the existence of the resulting OT map. Differentiability is required for implementing the training algorithm, since gradients of $A$ appear in the optimization process. This differentiability assumption was not explicitly stated in the original manuscript, and we added it in the revised version (Line 230). We appreciate the reviewer’s comment, which helped us improve the theoretical clarity of the paper.
>
> $ $
>
> ---
> > [Q2] Could the authors also comment on the cases where assumptions are not met in practice, but the method seems to work?
>
> **A.** A representative example is the **super-resolution task**. In this setting, we use a modified quadratic cost to match the resolution between low-resolution and high-resolution images:
> $$
> c_{q} (y,x) = \lVert Q(y) - x \rVert_{2}^{2}
> $$
> where $Q$ denotes the bicubic interpolation (Line 347). In this case, the **twist condition is not satisfied**. Hence, the existence of the OT map is not theoretically guaranteed. Nevertheless, as shown in Table 1, our **UOTIP works well on super-resolution in practice**. We attribute this to the local smoothing inductive bias of the generator network architectures, which is a variant of UNet. This inductive bias acts as a regularizer and stabilizes the transport even when the theoretical assumptions are not met.
>
> $ $
>
> ---
> > [Q3] In particular, does the framework extend to nonlinear or non-differentiable operators, or is it restricted to Lipschitz-continuous and differentiable forward models?
>
> **A.** Our framework extends to nonlinear forward operators, as demonstrated in the HDR reconstruction and nonlinear deblurring experiments in Table 1. However, the method cannot be applied to non-differentiable operators, since our training algorithm requires access to the gradients of $A$. In summary, UOTIP applies to forward models that are **Lipschitz-continuous and differentiable, regardless of whether they are linear or nonlinear**.
>
> $ $
>
> ---
> > [Q4] How would the method behave for inverse problems where A is discontinuous, piecewise-defined, or only approximately known? I believe a broader discussion of the characteristics of the inverse problems that could be addressed with the proposed method needs to be included in the paper.
>
> **A.** Thank you for raising the insightful question. Our current framework relies on the assumption that the forward operator $A$ is Lipschitz-continuous and differentiable, as these properties are needed both for (i) theoretical guarantees on the existence of the UOT map and (ii) practical gradient-based training. When $A$ is **discontinuous, piecewise-defined, or only approximately known**, these assumptions no longer hold, and the behavior of UOTIP becomes less predictable.
>
> If $A$ is piecewise-defined but differentiable almost everywhere, the method may still function in practice, similar to how it works for super-resolution, but we cannot provide theoretical guarantees. If $A$ is discontinuous or highly irregular, our model is not directly applicable, since gradients of $A$ are required during optimization. For the case where $A$ is only approximately known, UOTIP may retain some robustness, as suggested by our experiments on noise-type mismatch and multi-level noise, but this remains an open question that depends on the extent and structure of the approximation error.

---

### Official Review · Reviewer_84WS · 2025-10-31

**Soundness:** 2
**Presentation:** 3
**Contribution:** 2
**Rating:** 4
**Confidence:** 4

**Summary:**

This paper proposes UOTIP, a novel solver for unpaired inverse problems that learns an Unbalanced Optimal Transport (UOT) map from noisy measurements to clean signals with a standard likelihood-based cost function as guidance. The UOT framework provides robustness to multi-level noise and class imbalance, enabling UOTIP to achieve state-of-the-art performance on several linear and nonlinear inverse problem benchmarks.

**Strengths:**

1. The paper provides clear, empirical evidence for why UOT is superior to standard OT for inverse problem tasks.

2. Within its comparison class (other OT-based methods like NOT, OTUR, and RCOT), UOTIP achieves state-of-the-art performance across all four linear and nonlinear benchmarks.

3. The writing is clear and easy to follow.

**Weaknesses:**

1. The paper claims "state-of-the-art performance" , but the experimental comparison is limited to other Optimal Transport (OT) methods (NOT, OTUR, RCOT) . The actual state-of-the-art for many unpaired inverse problems often involves diffusion models or other advanced generative methods. Please compare with BlindDPS, GibbsDDRM, UNSB.

2. The paper's main likelihood cost $c_l$ is based on the $L_2$ norm, which is optimal only if you assume Gaussian measurement noise. The paper then tests this exact same Gaussian-derived cost function on Laplace noise and Poisson noise (Sec 4.2). It claims this shows "generalization", which is overclaimed. It just shows the model doesn't completely fail, not that it's robust. A solid paper would have derived the correct likelihood cost for Laplace (an $L_1$ norm) or Poisson noise to show the framework's flexibility, not just the robustness of a mismatched cost.

3. The paper presents the robustness to multi-level noise and class imbalance as key advantages of their model. However, the paper itself cites prior work for these exact properties of Unbalanced Optimal Transport.Therefore, the paper isn't discovering that UOT is robust; it's simply applying a known benefit of UOT to the inverse problem domain. The core novelty is just the application and the $\mathcal{A}$-dependent cost function, which is a standard approach used in [1][2].

4, This $c_q$ variant for 'real' unknown corruption setting was only tested on the two linear inverse problems. The paper provides no evidence for how this "blind" variant would perform on the much more complex nonlinear tasks (HDR and Nonlinear Deblurring).


5. Following my previous comment 4. The paper's claim of handling "unknown corruption operators" is misleading. The main contribution and study, the likelihood cost $c_l(y,x) = ||\mathcal{A}(x)-y||_2^2$, explicitly requires a known operator $\mathcal{A}$. The "blind" capability is relegated to a $c_q$-only variant that is not comprehensively evaluated. This variant was only tested on linear problems, providing no evidence for its performance on the more complex nonlinear tasks.

[1] Chung, H., Kim, J., Kim, S. and Ye, J.C., 2023. Parallel diffusion models of operator and image for blind inverse problems. In Proceedings of the IEEE/CVF Conference on Computer Vision and Pattern Recognition (pp. 6059-6069).

[2] Murata, N., Saito, K., Lai, C.H., Takida, Y., Uesaka, T., Mitsufuji, Y. and Ermon, S., 2023, July. Gibbsddrm: A partially collapsed gibbs sampler for solving blind inverse problems with denoising diffusion restoration. In International conference on machine learning (pp. 25501-25522). PMLR.

[3] Kim, B., Kwon, G., Kim, K. and Ye, J.C., Unpaired Image-to-Image Translation via Neural Schrödinger Bridge. In The Twelfth International Conference on Learning Representations.

**Questions:**

Same as my Weaknesses.

---

> ### Author Response · Authors · 2025-11-21
> **Response to Reviewer 84WS (1/3)**
>
> We sincerely thank the reviewer for carefully reading our manuscript and providing valuable feedback. We hope our responses to be helpful in addressing the reviewer's concerns. We highlighted the corresponding revisions in the manuscript in Green.
>
> $ $
>
> ---
> > [W1] The paper claims "state-of-the-art performance" , but the experimental comparison is limited to other Optimal Transport (OT) methods (NOT, OTUR, RCOT). The actual state-of-the-art for many unpaired inverse problems often involves diffusion models or other advanced generative methods. Please compare with BlindDPS, GibbsDDRM, UNSB.
>
> **A.** Thank you for raising this important point. We agree that diffusion-based inverse problem solvers such as BlindDPS, GibbsDDRM, and UNSB are strong baselines. However, these methods belong to a **fundamentally different modeling paradigm** from our work. Our method, UOTIP, is a **direct transport-map–based approach with NFE = 1**, whereas these approaches rely on iterative sampling with tens to hundreds of NFEs. Because of these core differences in generative mechanisms and inference complexity, we view these approaches as complementary rather than directly comparable. To avoid overstating our claims, we revised the manuscript to clarify this distinction and now state that UOTIP achieves **state-of-the-art performance among OT-based direct transport methods** (Line 64), which is the appropriate comparison category for our framework.
>
> $ $
>
> ---
>  > [W3] The paper presents the robustness to multi-level noise and class imbalance as key advantages of their model. However, the paper itself cites prior work for these exact properties of Unbalanced Optimal Transport.Therefore, the paper isn't discovering that UOT is robust; it's simply applying a known benefit of UOT to the inverse problem domain. The core novelty is just the application and the $A$-dependent cost function, which is a standard approach used in [1][2].
>
> **A.** We appreciate the reviewer’s thoughtful comment. It is true that robustness to class imbalance is a known theoretical property of Unbalanced OT (UOT). However, our contributions go beyond simply applying UOT to inverse problems.
>
> First, within the UOT literature, **UOT has not been explored in the inverse-problem setting**, and directly applying the vanilla formulation is not feasible. In inverse problems, observations are generated through a noisy forward operator $A$, and naively combining the likelihood-based cost with the standard UOT formulation generally **breaks the existence of an OT map** due to the ill-posed nature of the problem. Our method, UOTIP, is the first to **combine the informative likelihood cost with a regularizing quadratic term** (Remark 1) in order to both (i) guide the transport using measurement-model information and (ii) recover theoretical guarantees on map existence. This is fundamentally different from prior UOT-based generative modeling [1], where the quadratic cost has no practical meaning, or UOT-based point cloud completion [2], where an informative cost (infoCD) is used but without theoretical guarantees for the existence of a transport map.
>
> Moreover, while classical UOT theory discusses robustness under class imbalance, how this property manifests in inverse problems has never been studied. We reinterpret class-imbalance robustness in the inverse-problem context as robustness to **multi-level noise, which naturally occurs when measurements come from multiple sensors**. Our experiments verify that our model shows this robustness in practice.
>
> Second, within the inverse-problem community, **UOTIP is the first method to formulate inverse problems through Unbalanced OT**. This principled formulation leads to a practical one-step inverse problem solver with various beneficial properties such as robustness to multi-level noise, noise-type mismatch, and class imbalance. We also emphasize that **incorporating the likelihood cost into an OT or UOT map behaves fundamentally differently from MAP-based diffusion approaches** [3, 4]. MAP maximizes $\log p(x|y)$ for a single observation $y$, whereas the OT map $T$ with the likelihood cost maximizes the posterior **jointly over all observations** under the global constraint $T_{\sharp}\mu = \nu$. This global constraint is precisely what enables the robustness of UOTIP under the noise type mismatch in W2.
>
> [1] Choi, Jaemoo, Jaewoong Choi, and Myungjoo Kang. "Generative modeling through the semi-dual formulation of unbalanced optimal transport." NeurIPS 2023.
> [2] Lee, Taekyung, et al. "Unpaired Point Cloud Completion via Unbalanced Optimal Transport." ICML 2025.
> [3] Chung, Hyungjin, et al. "Parallel diffusion models of operator and image for blind inverse problems." CVPR 2023.
> [4] Murata, Naoki, et al. "Gibbsddrm: A partially collapsed gibbs sampler for solving blind inverse problems with denoising diffusion restoration." ICML 2023.

---

> ### Author Response · Authors · 2025-11-21
> **Response to Reviewer 84WS (2/3)**
>
> ---
> > [W2] The paper's main likelihood cost $c_l$ is based on the $L_2$ norm, which is optimal only if you assume Gaussian measurement noise. The paper then tests this exact same Gaussian-derived cost function on Laplace noise and Poisson noise (Sec 4.2). It claims this shows "generalization", which is overclaimed. It just shows the model doesn't completely fail, not that it's robust. A solid paper would have derived the correct likelihood cost for Laplace (an  $L_1$ norm) or Poisson noise to show the framework's flexibility, not just the robustness of a mismatched cost.
>
> **A.** We appreciate the reviewer for the insightful comment. Following the reviewer’s suggestion, we conducted **additional experiments using the correct likelihood cost functions** for Laplace and Poisson noise. For Laplace noise, we used the L1 likelihood:
> $$c_{l, laplace}(\mathbf{y}, \mathbf{x}) = -\lVert \mathbf{y} - A \mathbf{x} \rVert_{1} $$
>
> For Poisson noise, we used the standard scaled quadratic approximation (as in [1]):
>
> $$c_{l, poisson}(\mathbf{y}, \mathbf{x}) =- \lVert  \mathbf{y} - A(\mathbf{x}) \rVert_{ \Lambda }^{2} \qquad   {[\Lambda]}_{ii} \triangleq \frac{1}{2 y_i}, \quad  \text{where} \quad \lVert \mathbf{a} \rVert _{ \Lambda }^{2} \triangleq \mathbf{a}^{T}  \Lambda \mathbf{a}.$$
>
>
> The results are as follows:
>
> **Gaussian Deblur**
>
> | Ratio          | Method                      | PSNR ↑ | SSIM ↑  | LPIPS ↓ | FID ↓   |
> |:--|--|--|--|--|--|
> | **Laplace noise** | NOT                        | 19.50  | 0.5810  | 0.234   | 57.921  |
> |                | OTUR                        | 23.63  | 0.7074  | 0.137   | _23.825_ |
> |                | **UOTIP (Ours)**            | _23.98_ | _0.7153_ | _0.123_ | **21.846** |
> |                | UOTIP (Ours-Laplace likelihood)        | **24.12** | **0.7204** | **0.121** | 25.342       |
> | **Poisson noise** | NOT                        | 19.15  | 0.5584  | 0.246   | 55.368  |
> |                | OTUR                        | _23.14_ | _0.6919_ | _0.149_ | _25.840_ |
> |                | **UOTIP (Ours)**            | **23.47** | **0.6938** | **0.136** | **23.669** |
> |                | UOTIP (Ours-Poisson likelihood)        | 18.06| 0.4295      | 0.403 | 290.053       |
>
>
> **Super Resolution 4x**
>
> | Ratio          | Method                      | PSNR ↑ | SSIM ↑  | LPIPS ↓ | FID ↓   |
> |:--|--|--|--|--|--|
> | **Laplace noise** | NOT                        | 20.41  | 0.6277  | 0.208   | 50.228  |
> |                | OTUR                        | 23.72  | 0.7252  | 0.130   | _23.931_ |
> |                | **UOTIP (Ours)**            | **24.04** | **0.7350** | **0.118** | **19.629** |
> |                | UOTIP (Ours-Laplace likelihood)        | _23.99_ | _0.7317_ | _0.122_ | 27.053       |
> | **Poisson noise** | NOT                        | 19.28  | 0.5820  | 0.248   | 57.606  |
> |                | OTUR                        | _23.26_ | _0.7020_ | _0.148_ | _26.222_ |
> |                | **UOTIP (Ours)**            | **23.59** | **0.7163** | **0.136** | **20.396** |
> |                | UOTIP (Ours-Poisson likelihood)        | 22.56      | 	0.6075       | 0.351       | 175.036      |
>
> Under **Laplace noise**, UOTIP with the Laplace likelihood performs better or comparable with the Gaussian-likelihood version, demonstrating that the framework naturally supports the correct likelihood. Interestingly, under **Poisson noise**, using the Poisson likelihood yields worse performance than using the Gaussian likelihood. This aligns with known numerical instabilities in Poisson likelihood objectives (Appendix C.4 of [1]). Importantly, UOTIP with Gaussian likelihood remains stable and competitive, illustrating practical robustness even under likelihood mismatch.
>
> The goal of the experiment in Sec. 4.2 was to **evaluate robustness under noise-type mismatch**, which is a **practically important setting where the true noise distribution is often unknown**. In this scenario, the question is not whether the cost matches the noise exactly, but whether the method is robust under noise-type mismatch. Our results on Laplace and Poisson noise (Fig. 4(a), Table 7) show that UOTIP is significantly **more stable and robust than the baselines, even when the likelihood is misspecified**. This setting reflects many real-world situations where the measurement noise type cannot be reliably predetermined. To avoid confusion, we revised the sentence in Line 452 from:
>
> "we examine whether the proposed model can generalize..."
>
> to:
>
> "we examine the robustness of our model under noise-type mismatch by testing on alternative noise distributions while keeping the likelihood cost $c_l$ fixed as Gaussian."
>
> This wording more accurately reflects the intention of the experiment.
>
> [1] Chung, Hyungjin, et al. "Diffusion Posterior Sampling for General Noisy Inverse Problems." The Eleventh International Conference on Learning Representations.

---

> ### Author Response · Authors · 2025-11-21
> **Response to Reviewer 84WS (3/3)**
>
> ---
> > [W4] This $c_q$ variant for 'real' unknown corruption setting was only tested on the two linear inverse problems. The paper provides no evidence for how this "blind" variant would perform on the much more complex nonlinear tasks (HDR and Nonlinear Deblurring).
>
> **A.** Our quadratic-cost variant $c_{q}$  was introduced as an ablation study on the choice of cost function. From the UOT formulation, this quadratic-only variant $c_{q}$ naturally shows potential as a **blind corruption solver only in settings where the measurement and signal share a similar structure**, such as Gaussian deblurring and super-resolution. For more complex nonlinear tasks (e.g., HDR and nonlinear deblurring), this structural similarity does not hold. Therefore, we do not position $c_{q}$  as a universal blind solver, but rather as a proof of concept for cases where the forward operator is close to identity.
>
> We would like to note that **our claim regarding blind corruption is explicitly limited to the linear inverse problem case** in the Abstract and Contribution summary (Lines 77). To avoid confusion, we clarified this point in the revised manuscript (Lines 473) as follows:
> > This suggests that our quadratic-cost-only formulation has potential as an effective blind inverse problem solver when the corruption operator $A$ preserves the signal structure, such as in Gaussian deblurring.
>
> $ $
>
> ---
> > [W5] Following my previous comment 4. The paper's claim of handling "unknown corruption operators" is misleading. The main contribution and study, the likelihood cost $c_l(y,x) = ||A(x)-y||_2^2$, explicitly requires a known operator $A$. The "blind" capability is relegated to a $c_q$-only variant that is not comprehensively evaluated. This variant was only tested on linear problems, providing no evidence for its performance on the more complex nonlinear tasks.
>
> **A.** As noted in the response to W4, we expect the quadratic-cost variant $c_{q}$ to exhibit feasibility in blind setting only when the corruption operator approximately preserves signal structure. However, blind operator case is not the primary goal of this work. The main contribution of the paper is a **principled UOT-based formulation for inverse problems** that remains **robust under noise-type mismatch, class imbalance, and multi-level observation noise**. For example, even when trained with a Gaussian likelihood, UOTIP exhibits strong robustness under Laplace and Poisson noise, demonstrating that our method remains stable even when the corruption model is partially misspecified. This robustness is particularly important in real-world applications where noise characteristics cannot always be reliably specified.

---

### Official Review · Reviewer_8LLr · 2025-11-01

**Soundness:** 3
**Presentation:** 3
**Contribution:** 3
**Rating:** 4
**Confidence:** 4

**Summary:**

This paper aims to recover a clean target signal distribution from a noisy observation distribution within an 'unpaired' inverse problem framework.  In particular, the authors propose learning an Unbalanced Optimal Transport (UOT) mapping to transform the observed distribution into the target distribution.  The authors finally propose training a hybrid loss and demonstrate its effectiveness through experimental results on both linear and nonlinear inverse problems (natural image restorations).

**Strengths:**

- The idea of applying UOT to unpaired inverse problems is very interesting. Since UOT can address distributional mismatch issues, it could provide an alternative solution to problems that standard OT struggles to handle directly.

- The authors also studied the theoretical conditions for the existence of OT maps (twist and semi-concavity) and tried to demonstrate the rigour of cost design in satisfying solubility conditions, thereby surpassing some empirical methods.

- The experimental results are good with respect to multiple tasks and noise types/levels. It shows that better results are achieved when the noise model aligns with the cost.

**Weaknesses:**

- Theoretical proof can be improved. For example, the paper uses Fathi and Figalli's theorem to guarantee the existence of the OT map. However, no rigorous proof is provided as to whether the conditions within that theorem (e.g. local semi-concavity, left twist and the zero-mass condition) are met in high-dimensional image spaces and under the cost function, c(y, x).

- The claim that adding a quadratic term guarantees the existence and injectivity of a map depends on key constants (e.g. λ > Lipschitz L). How can L be estimated in practice? Does the proof rely on hidden strong assumptions?

- No presented convergence guarantees or training dynamics plots weaken confidence in practicality.

- The experimental results are not significantly superior.

- The writing could be further improved. Typo: in Eq. (23), the last term should be written as $A(x)$ instead of $A(y)$.

**Questions:**

- What network architectures are used for training? It's necessary to complete hyperparameter descriptions, seeds, training times and memory consumption are needed.

- Are the baselines being trained and tuned equivalently? There are no comparisons to diffusion/posterior-sampling methods.

- What if the inverse problem is that x and y are in different domains? For example, consider MRI image reconstruction. I am curious about the performance on e.g. biomedical imaging applications.

- Eq(17)-Eq(20): What if the noise models/levels are unknown?

- Verifying that c(y, x) satisfies the Fathi and Figalli assumptions (local semi-concavity, left twist and μ not assigning mass to certain sets) is necessary for the task at hand.

---

> ### Author Response · Authors · 2025-11-21
> **Response to Reviewer 8LLr (1/3)**
>
> We sincerely thank the reviewer for carefully reading our manuscript and providing valuable feedback. We hope our responses to be helpful in addressing the reviewer's concerns. We highlighted the corresponding revisions in the manuscript in Blue.
>
> $ $
>
> ---
> > [W1] Theoretical proof can be improved. For example, the paper uses Fathi and Figalli's theorem to guarantee the existence of the OT map. However, no rigorous proof is provided as to whether the conditions within that theorem (e.g. local semi-concavity, left twist and the zero-mass condition) are met in high-dimensional image spaces and under the cost function, c(y, x).
>
> > [Q5] Verifying that c(y, x) satisfies the Fathi and Figalli assumptions (local semi-concavity, left twist and μ not assigning mass to certain sets) is necessary for the task at hand.
>
>
> **A.** We appreciate the reviewer for the careful comment. As discussed in Remark 1 and Appendix C, **our cost function satisfies the conditions of Fathi and Figalli's theorem (local semi-concavity and left twist condition)** for all tasks except for the super-resolution. In the super-resolution task, the quadratic cost is modified to match the high-resolution and low-resolution images (Line 348). We revised the manuscript to clarify that this modification lies outside the theorem’s condition and should be viewed as a practical extension rather than a theoretically guaranteed case.
>
> Regarding the zero-mass condition, **this requirement is met under our assumption that the source measure is absolutely continuous** with respect to the Lebesgue measure (Line 83). This assumption is satisfied if the observation noise on the forward operator follows a Gaussian distribution, because Gaussian smoothing makes the resulting measure absolutely continuous. Note that the absolute continuity is a standard assumption in the Neural Optimal Transport approaches, such as [1, 2, 3]. Under this assumption, Fathi and Figalli's theorem applies to our formulation.
>
> [1] Makkuva, Ashok, et al. "Optimal transport mapping via input convex neural networks." ICML 2020.
> [2] Rout, Litu, Alexander Korotin, and Evgeny Burnaev. "Generative modeling with optimal transport maps." ICLR 2022.
> [3] Choi, Jaemoo, Jaewoong Choi, and Myungjoo Kang. "Generative modeling through the semi-dual formulation of unbalanced optimal transport." NeurIPS 2023.
>
> $ $
>
> ---
> > [W2] The claim that adding a quadratic term guarantees the existence and injectivity of a map depends on key constants (e.g. λ > Lipschitz L). How can L be estimated in practice? Does the proof rely on hidden strong assumptions?
>
> **A.** We appreciate the reviewer for the thorough comment. In practice, we do not directly estimate the Lipschitz constant $L$ nor explicitly tune $\lambda$. Instead, the role of $\lambda$ in the theoretical analysis is reflected in the **practical hyperparameter $\tau$ used in our cost function (Eq. 12)**, which controls the intensity of the cost function. While this provides a stable approach in practice, we agree that developing a more systematic procedure for estimating or adapting $\lambda$ would be an interesting direction for future research. **Our Remark 1 is intended to justify the additional quadratic term at a theoretical level by showing that, under suitable conditions, such a term ensures existence and injectivity of the transport.** This existence guarantee can be interpreted as addressing the ill-posedness challenge for the inverse problems.
>
> $ $
>
> ---
> > [W3] No presented convergence guarantees or training dynamics plots weaken confidence in practicality.
>
> **A.** We thank the reviewer for the suggestion to strengthen the practical confidence in our model. In the revised manuscript, we added training dynamics plots (performance metrics vs. total iterations) in Fig. 6. As shown, **our model exhibits stable training behavior: a rapid improvement in the early stages, followed by steady, gradual gains throughout training**. These results further support the practicality and reliability of our approach.

---

> ### Author Response · Authors · 2025-11-21
> **Response to Reviewer 8LLr (2/3)**
>
> ---
> > [W4] The experimental results are not significantly superior.
>
> **A.** We respectfully disagree with the assessment that the improvements are not significant. Across all benchmarks, **our method demonstrates substantial and consistent gains, achieving the best performance on nearly all evaluation metrics**. In particular, the improvements in FID are large and meaningful. For example, in Gaussian deblurring on FFHQ, FID improves from 24.3 → 21.2, and on AFHQ from 30.8 → 12.6, representing a substantial improvement in generation quality.
>
> Moreover, beyond improvements on standard benchmarks, our analysis highlights the **unique strengths of UOTIP based on the Unbalanced Optimal Transport formulation**. In particular, UOTIP handles multi-level observation noise (Tables 2 and 4) and class imbalance (Fig. 3 and Table 5) more effectively than competing baselines. These results demonstrate not only quantitative improvements but also improved robustness across challenging and practically relevant scenarios.
>
> $ $
>
> ---
> > [W5] The writing could be further improved. Typo: in Eq. (23), the last term should be written as $A(x)$ instead of $A(y)$.
>
> **A.** Thank you for the careful advice. We corrected Eq. (23) and highlighted the correction in Blue in the revised manuscript.
>
> $ $
>
> ---
> > [Q1] What network architectures are used for training? It's necessary to complete hyperparameter descriptions, seeds, training times and memory consumption are needed.
>
> > [Q2] Are the baselines being trained and tuned equivalently? There are no comparisons to diffusion/posterior-sampling methods.
>
> **A.** As described in **Appendix B (Implementation details)**, we strictly followed the original configurations for the two baselines (OTUR and RCOT). For the NOT model, we employed the stronger and larger UNet-based generator (a slight variant of NCSN++) and the potential network employed in UOTM. For our model (UOTIP), we adopted **the same generator architecture as OTUR** and **the same potential network architecture as UOTM [1]** to ensure architectural fairness. All other training hyperparameters, including learning rate, the cost intensity hyperparameter $\tau$, batch size, and the convex conjugate $\psi^{*}$, are listed in Appendix B.
>
> Following the reviewer's suggestion, we additionally included a comparison of **training time** and **parameter count** (as a proxy for memory consumption) in the Efficiency Analysis paragraph of the revised manuscript. OTUR is the most efficient baseline, and our UOTIP is the second most efficient. These results verify that all experiments were conducted under **equivalent and fair conditions**. The detailed results are as follows:
>
> |Training (60000 iteration)| Time (sec)|
> |:--- |:---|
> |NOT| 64,763.4|
> |RCOT| 117,126.1|
> |OTUR| 24,255.6 |
> |Ours (UOTIP)| 52,326.2 |
>
> | Memory consumption | Number of Parameters (M)|
> |:--- |:---|
> |NOT| 65.934 |
> |RCOT| 77.442 |
> |OTUR| 20.369 |
> |Ours (UOTIP)| 31.474 |
>
> Lastly, we agree with the reviewer that diffusion/posterior-sampling inverse problem solvers represent an important class of methods. However, these approaches follow a fundamentally different modeling paradigm with iterative sampling procedures, whereas UOTIP is a direct transport method (with NFE 1) grounded in Unbalanced Optimal Transport. Because the two paradigms differ fundamentally in inference mechanism, training objective, and computational cost, we consider them complementary rather than directly comparable.
>
> [1] Choi, Jaemoo, Jaewoong Choi, and Myungjoo Kang. "Generative modeling through the semi-dual formulation of unbalanced optimal transport." NeurIPS 2023.
>
> $ $
>
> ---
> > [Q3] What if the inverse problem is that x and y are in different domains? For example, consider MRI image reconstruction. I am curious about the performance on e.g. biomedical imaging applications.
>
>
> **A.** Thank you for raising this important question. **UOTIP is applicable even when $x$ and $y$ lie in different domains**, unlike standard quadratic-cost OT baselines (e.g., NOT). Our method uses the likelihood cost $c(y,x) = \| A(x) - y \|_{2}^{2}$. Therefore, the comparison is always conducted in the measurement domain of $y$ via the forward operator $A$. This makes the framework naturally compatible with inverse problems where measurements live in a transformed or lower-dimensional space. For example, in the suggested MRI image reconstruction, the measurement $y$ is obtained in the frequency domain through the Fourier transform and subsampling operator [1]. Although biomedical imaging applications such as MRI were not investigated in this paper, we agree that extending UOTIP to these applications would be a promising direction for future work.
>
> [1] Zheng, Hongkai, et al. "Inversebench: Benchmarking plug-and-play diffusion priors for inverse problems in physical sciences." ICLR 2025.

---

> ### Author Response · Authors · 2025-11-21
> **Response to Reviewer 8LLr (3/3)**
>
> > [Q4] Eq(17)-Eq(20): What if the noise models/levels are unknown?
>
>
> **A.** Thank you for the insightful question. We evaluated the unknown-noise scenario through two experimental settings.
>
> First, as shown in Lines 450 - 463, we tested whether our model, trained with a Gaussian likelihood, can generalize to **different noise types**, such as Laplace and Poisson noise (Eqs. 21–22). Fig. 4(a) and Table 6 demonstrate that our method is **substantially more robust** than baseline methods under these mismatched noise distributions.
>
> Second, we evaluated our model under **multi-level observation noise**, which is a more challenging setting than the (unknown) single noise level, as described in Lines 403–413. Specifically, the observation noise is sampled from a **Gaussian mixture**. This setting naturally arises when measurements come from multiple sensors. Even in this heterogeneous-noise regime, our model achieves competitive or superior performance across almost all metrics, outperforming other baselines.
>
> Finally, even in the single-noise-level setting, our method does not explicitly require the ground-truth noise level. Instead, the model adapts implicitly through the noise-intensity parameter $\tau$ in Eq. 12. These results demonstrate that UOTIP remains effective when the noise model or noise level is unknown

---

### Author Response · Authors · 2025-12-01
**Summary of Rebuttal Progress and Addressed Reviewer Concerns (2/2)**

---

**4. Reviewer 8dHT:  Novelty and Assumptions on the Forward Operator**

The major concerns from Reviewer 8dHT are as follows:
- **[W1 - W3] Novelty vs. prior work**
    - We clarified that no prior work integrates OT maps, UOT, and likelihood-based costs for unpaired inverse problems. Although each component exists individually, UOTIP is the first principled and theoretically justified integration of these ideas for unpaired inverse problems, offering new theoretical existence guarantees and robustness properties unavailable in prior work.

- **[Q1-4] Assumptions on the forward operator $A$ that are required for our theoretical results.**
    - We clarified that our theory assumes $A$ is Lipschitz continuous and differentiable, and we explained how these assumptions enter the OT-map existence proof and the training procedure.

$ $

---
**5. Reviewer nmHQ: Realism of the Setting, Comparisons, Presentation**

The major concerns raised by Reviewer nmHQ focus on the **realism of the task setting**, including [W3] the blind inverse problem scenario, [W7, Q3] the role of class imbalance in inverse problems, and [W8] the feasibility of an unpaired setting even when the operator is known. The minor concerns include [W1] discussions of related work outside OT, and [W4, W5, W6, Q4] clarity and presentation of the paper.

- **[W3, W7, Q3, W8] Realistic task setting**
    - We clarified that each of our task settings is realistic and provided real-world examples. For example, in MRI reconstruction, paired data are generally unavailable even when the forward operator is known. Therefore, this is an example of the unpaired inverse problem with a known forward operator. Moreover, following the reviewer’s suggestion, we conducted new experiments using synthetically generated paired data based on the known forward operator. This experiment showed that our model further improves when supervised loss terms are incorporated. These results show that UOTIP integrates supervised information smoothly and benefits from paired augmentation when available.
- **[W1] Discussion outside OT**
    - We conducted additional comparisons to a GAN-based baseline (OT-cycleGAN), included the discussion on learned regularizer methods in the related works, and clarified why diffusion/PnP priors belong to a fundamentally different category from UOTIP. In the additional experiments, UOTIP outperformed OT-cycleGAN.
- **[W4, W5, W6, Q4] Paper presentation**
    - We revised the manuscript to include training algorithms, extended operator/noise descriptions, and enhanced experimental details.

These concerns were fully addressed via additional experiments, clarifications, and manuscript improvements.

$ $

---

Given that reviewers were unable to update their evaluations after our detailed responses, we respectfully ask the Area Chair to consider the content and outcomes of the rebuttal, rather than the reverted initial scores. We believe the additional experiments, clarifications, and theoretical justifications show that all major concerns have been comprehensively addressed, and that the paper meets the standards of ICLR.

Sincerely,
Authors

---

### Author Response · Authors · 2025-12-01
**Summary of Rebuttal Progress and Addressed Reviewer Concerns (1/2)**

Dear Area Chair,

We understand that the recent technical issues with OpenReview have led to the substitution of the Area Chair and a freezing of review scores. We would like to express our gratitude to the new AC for taking on this responsibility under challenging circumstances.

During the rebuttal, we carefully addressed all reviewers’ concerns and substantially improved the manuscript. Since the discussion phase is frozen, we provide a brief summary of how reviewers’ major concerns were addressed in the rebuttal.

$ $

---
**1. Core Contribution Recap**

Our manuscript proposes a new framework for solving inverse problems in an unpaired setting by leveraging Unbalanced Optimal Transport (UOT). By integrating a likelihood-based cost with a quadratic cost term, we provide theoretical guarantees for the existence of an OT map as an inverse problem solver. Empirically, our method demonstrates strong performance across diverse inverse problems. In particular, our method is robust to multi-level noise, heterogeneous noise, class imbalance, different noise types, and certain blind-operator scenarios, without requiring paired training data.

$ $

---
**2. Reviewer 8LLr: Theoretical Clarifications, Training Details**

Reviewer 8LLr focused primarily on theoretical validity and implementation details. We addressed these concerns as follows:

- **[W1, Q5]  Applicability of the Fathi–Figalli conditions for ensuring OT map existence**
   - We clarified precisely when the theorem’s conditions hold. Specifically, our cost function and probability measures satisfy these conditions. Also, we identified that the super-resolution setting is a practical extension beyond the theorem’s assumptions and revised the manuscript accordingly.

- **[W2] Practical implementation of a cost function designed to satisfy the twist condition**
    - The reviewer asked how the theoretically required conditions relate to implementation. We clarified that the theory requires the cost to be dependent on the Lipschitz constant of the forward operator, whereas in practice, this corresponds to tuning a hyperparameter that controls the strength of the quadratic cost.

- **[W3, Q1] Training dynamics, training time, memory consumption.**
   - We added a detailed report on training dynamics, training time, and parameter counts in the revised version of the manuscript.

$ $

---
**3. Reviewer 84WS:  Likelihood Costs, Baselines, Novelty**

The major concerns from Reviewer 84WS are as follows:

- **[W2] Requested experiments with non-Gaussian likelihood costs**
    - We added new experiments using Laplace and Poisson likelihood costs. The results show that our UOTIP model is generalizable with Laplace likelihood cost. Moreover, we clarified that the goal of our original expeirments was to evaluate robustness under noise-type mismatch, which is a practically important setting where the true noise distribution is often unknown. Therefore, we claimed that the original noise-type mismatch setup is also appropriate for evaluating the robustness of our framework.
- **[W1] Request for comparison with diffusion-based solvers**
    - We clarified that diffusion-based solvers are fundamentally different (iterative sampling, tens–hundreds of NFEs), whereas UOTIP is a direct transport method (NFE = 1); thus, they are not directly comparable baselines.
- **[W3] Novelty of combining UOT and likelihood terms**
    - We clarified that our contribution is not a simple concatenation of known ideas; UOTIP is the first UOT-based inverse problem solver that guarantees OT-map existence. Moreover, our contribution includes the reinterpretation of class imbalance within the inverse problem context as robustness to multi-level noise, which naturally occurs when measurements come from multiple sensors.

These concerns were fully addressed in the rebuttal with additional experiments and clarification of the novelty of our model.

---

### Meta-Review · Area_Chair_mrTw · 2025-12-21

**Summary:**

This paper applies Unbalanced Optimal Transport (UOT) to solve unpaired image inverse problems. The reviewers raised several concerns in the first round of review. Major concerns include the novelty of the approach: while UOT itself is not new and has been studied extensively, this work applies it to inverse problems for the first time, but reviewers questioned whether this application alone constitutes sufficient novelty. In particular, one reviewer pointed out that the unpaired setting assumed in this paper may not be common in inverse problems when the forward operator is known.

Reviewers also expressed concerns about the experimental evaluation. The numerical comparisons are mainly conducted against GAN-based methods and do not include diffusion-based approaches, which are widely regarded as stronger baselines for inverse problems. Additional concerns include the use of a Gaussian likelihood to handle other types of noise and a lack of clarity in parts of the theoretical proof.

During the rebuttal, the authors emphasized that the primary contribution lies in applying existing UOT formulations to inverse problems and in demonstrating that favorable properties of UOT, such as robustness, are preserved in this setting. While some concerns were clarified, the reviewers’ critiques regarding novelty and baseline comparisons remain major issues and were not fully resolved. As a result, it is unlikely that reviewers would substantially increase their scores. Given these unresolved concerns, I do not recommend acceptance.

**Reviewer Concerns:**

Addressed Concerns
1. Comparisons with GAN-based methods
The authors added new experiments showing that the proposed method outperforms existing GAN-based baselines. These results are consistent and help strengthen the empirical evaluation within this comparison class.
2. Use of appropriate likelihoods for Laplace and Poisson noise
The authors added experiments using Laplace and Poisson likelihood costs for the corresponding noise models. The results are meaningful and consistent with the overall empirical findings.
3/ Clarity of writing and presentation
The revised manuscript is greatly improved in clarity, with better explanations of the method, assumptions, and experimental setup.

Outstanding (Main) Concerns
1. Lack of comparison with diffusion-based methods
One reviewer suggested including diffusion-based methods as stronger baselines. The authors argue that diffusion models require many NFEs and are therefore not directly comparable to GAN-based or one-step methods. While the difference in inference cost is valid, it would be more informative to include diffusion-based methods together with a runtime or NFE comparison, rather than omitting them entirely. The absence of such comparisons makes it difficult to properly position the proposed method relative to the state of the art.
2. Realism of the unpaired training data assumption
One reviewer questioned whether the unpaired setting is realistic for inverse problems when the forward operator is known, since paired data can be easily generated by applying the forward operator to known ground-truth images. In the rebuttal, the authors argued that MRI applications often involve unpaired data and showed that incorporating paired data can further improve performance. However, even in MRI settings, the reviewer’s concern remains largely valid: for data where ground-truth images are available, the corresponding measurements can be readily generated, allowing methods that rely on paired data to be applied. A comparison with such paired-data methods, ideally on a real (even small-scale) MRI dataset, is still missing.

**Reviewer Scores:**

Reviewer 8LLr (score 4): May increase the score slightly (e.g., from 4 to 5), as concerns about clarity and implementation details have largely been addressed.

Reviewer 84WS (score 4): Likely to maintain the current score, as the main concern regarding the lack of comparison with diffusion-based baselines remains unresolved.

Reviewer 8dhT (score 6): Likely to maintain the current score, as the concern about novelty, particularly that the core component (UOT) already exists outside the inverse problem setting, remains valid.

Reviewer nmHQ (score 2): Likely to maintain the current score or increase it slightly, as the primary concern regarding the realism of the unpaired setting in inverse problems has not been fully justified.

---

### Decision · Program_Chairs · 2026-01-26

Reject